# Legume rhizodeposition promotes nitrogen fixation by soil microbiota under crop diversification

Mengjie Qiao [1,2,10], Ruibo Sun[3,10], Zixuan Wang[1,4], Kenneth Dumack [5], Xingguang Xie[6], Chuanchao Dai [7], Ertao Wang [6], Jizhong Zhou [8], Bo Sun [1,11], Xinhua Peng[1], Michael Bonkowski [5,9] & Yan Chen [1] ✉

Biological nitrogen fixation by free-living bacteria and rhizobial symbiosis with legumes plays a key role in sustainable crop production. Here, we study how different crop combinations influence the interaction between peanut plants and their rhizosphere microbiota via metabolite deposition and functional responses of free-living and symbiotic nitrogen-fixing bacteria. Based on a long-term (8 year) diversified cropping field experiment, we find that peanut co-cultured with maize and oilseed rape lead to specific changes in peanut rhizosphere metabolite profiles and bacterial functions and nodulation. Flavonoids and coumarins accumulate due to the activation of phenylpropanoid biosynthesis pathways in peanuts. These changes enhance the growth and nitrogen fixation activity of free-living bacterial isolates, and root nodulation by symbiotic *Bradyrhizobium* isolates. Peanut plant root metabolites interact with *Bradyrhizobium* isolates contributing to initiate nodulation. Our findings demonstrate that tailored intercropping could be used to improve soil nitrogen availability through changes in the rhizosphere microbiome and its functions.

Chemical signaling between plants and soil microbiota plays a critical role in microbial symbioses and rhizosphere microbiome assembly[1,2]. The secondary metabolites exuded by plant roots are believed to attract and filter species-specific microbial taxa[3,4], including microbiota that complement their host's functional repertoire with traits not encoded in the plant genome[5], such as biological nitrogen fixation and phosphorus uptake[6,7]. In turn, compounds released by rhizosphere microbes trigger plant responses that further adjust microbiome specificity and composition[8,9]. This continuous chemical dialog is reflected in the metabolic deposition of the host plant rhizosphere, also known as rhizodeposition[10–12].

Although great mechanistic insights have been obtained on rhizosphere chemical signaling and rhizomicrobiome assembly of individual plant species[1,13,14], much less is known about how these processes are influenced by interspecific interactions between coexisting plant species. Various studies have found that interspecific neighbor-driven species recognition can induce a metabolic response in the neighbor and change the chemical composition of its

¹State Key Laboratory of Soil and Sustainable Agriculture, Institute of Soil Science, Chinese Academy of Sciences, Nanjing 210008, China. ²University of Chinese Academy of Sciences, Beijing 100049, China. ³Anhui Province Key Lab of Farmland Ecological Conservation and Nutrient Utilization, College of Resources and Environment, Anhui Agricultural University, Hefei 230036, China. ⁴College of Land Resource and Environment, Jiangxi Agricultural University, Nanchang 330045, China. ⁵Terrestrial Ecology, Institute of Zoology, University of Cologne, Zülpicher Str 47b, Cologne 50674, Germany. ⁶National Key Laboratory of Plant Molecular Genetics, Center for Excellence in Molecular Plant Sciences, Institute of Plant Physiology and Ecology, Chinese Academy of Sciences, Shanghai 200032, China. ⁷College of Life Sciences, Nanjing Normal University, Nanjing 210023, China. ⁸Institute for Environmental Genomics and Department of Microbiology and Plant Biology, University of Oklahoma, Norman 73019, USA. ⁹Cluster of Excellence on Plant Sciences (CEPLAS), University of Cologne, Cologne 50674, Germany. ¹⁰These authors contributed equally: Mengjie Qiao, Ruibo Sun. ¹¹Deceased: Bo Sun. ✉e-mail: chenyan@issas.ac.cn

rhizosphere[15–17]. Such chemical alterations theoretically drive subsequent changes in the structure and function of the rhizomicrobiome. However, recent field studies on chemical feedbacks between plant species have focused more on non-kin species defense[18–20], and less on how these chemical cues may alter the rhizomicrobiome and microbially mediated functions that affect the plant fitness of the species involved[21,22].

The question of how interspecific effects on plant fitness are shaped by rhizomicrobiome feedbacks is particularly relevant in the context of diversified cropping systems, in which crop species diversity is increased in space (e.g. intercropping) and time (crop rotation). Field studies on such systems often demonstrate improved performance of key food crops, especially in intercropping systems including legumes[23]. One of the keys to the success in these systems is improved nitrogen (N) availability through biological $N_2$ fixation, both by free-living bacteria and rhizobial symbiosis with legumes. The latter, in particular, requires finely tuned reciprocal signal transduction systems as the host plant has to reprogram root growth and invest in nodule structures before any gain from the symbiosis is measurable[24]. These processes are likely to be influenced by the other (non-legume) crops in the system (see e.g. Li et al.[21]). By identifying specific rhizosphere metabolites that modulate legume rhizomicrobiome assembly and biological $N_2$ fixation in these systems, we are able to gain detailed mechanistic insights into plant-rhizosphere feedback processes and understand why some crop combinations work better than others.

To gain such mechanistic insights, we combined biochemistry, molecular biology and crop ecology to investigate how the chemical signal exchange between legumes and their rhizosphere microbiota is influenced by different crop combinations, and how this in turn affects biological $N_2$ fixation and legume crop performance. As a model for legumes we focused on peanut, which is widely grown in the tropics and subtropics and has edible seeds that develop underground. For this analysis we gathered field data from an eight-year-old crop diversification experiment (Fig. S1), in which peanut was grown in monoculture (PP), in rotation with oilseed rape (P-R), and in an intercropping system with maize, rotated with oilseed rape (PM-R). Using multiomics analysis (including nontargeted metabolomics and transcriptomics of peanut roots), bacterial isolation and bacterial inoculation experiments, we aimed to (i) identify metabolic cues in the peanut rhizosphere that are influenced by the other crop species in the system, and (ii) to investigate whether and how these metabolites trigger rhizosphere microbiota and specific functions, biological $N_2$ fixation in particular. Our results provide a mechanistic link between plant diversity and belowground functioning and illustrate how the chemical dialogs between plants and their rhizomicrobiome result in a mutual plant-microbe alliance that improves fitness of both.

## Results
### Crop diversification enhances peanut production, root nodulation and free-living $N_2$ fixation
Crop diversification had a significant positive effect on peanut performance. In the system with highest crop diversity, i.e. where peanut was intercropped with maize and rotated with oilseed rape, average peanut height, biomass and fruit weight were increased by at least 19%, 66% and 46%, respectively, compared with peanut monoculture and peanut-rape rotation ($p < 0.05$, Turkey-HSD; Fig. S2). It finally resulted in 51% higher nitrogen uptake of peanut in crop mixture than in the other two crop systems ($p < 0.05$, Turkey-HSD). Although peanut biomass was lower in the rotation system (P-R) than in the monoculture system ($p < 0.05$, Turkey-HSD), this did not affect fruit yield ($p > 0.05$, Turkey-HSD). In terms of root symbiotic $N_2$ fixation, we investigated root nodulation. The differences between treatments were striking (Fig. 1a, b): peanut roots in crop mixture had a three-fold higher nodule density ($p < 0.001$, Turkey-HSD) and a six-fold higher nodule-to-root mass ratio ($p < 0.001$, Turkey-HSD) than peanuts grown in

monoculture. When comparing the two rotation systems with and without maize intercropping (PM-R versus P-R), peanut nodule density was 50% higher and nodule-to-root mass ratio was twice as high in crop mixture compared to peanut-rape rotation ($p < 0.05$, Turkey-HSD). Linear regressions showed that peanut plant biomass (Fig. 1d, e; $p = 0.026$, t-test) were significantly positively correlated with nodule-to-root mass ratio, while rhizosphere ammonium levels (Fig. 1g, h; $p \leq 0.005$, t-test) were positively correlated significantly with both nodule density and nodule-to-root mass ratio. With respect to the rhizosphere free living $N_2$ fixation, microbial immobilization of molecular $^{15}N$ was measured using inoculation of $^{15}N_2$ isotope labeling. Obviously, soil $\delta^{15}N$ in crop mixture was the highest, with 16% and 4% higher ($p < 0.05$, Turkey-HSD) than peanut monoculture and peanut-rape rotation after 7 days of incubation, respectively (Fig. 1c). Peanut rhizosphere ammonium and nitrate nitrogen levels were positively correlated with soil $^{15}N$ fixation (Fig. 1f and i; $p \leq 0.013$).

Consistent with the observed treatment effects on peanut growth and nitrogen fixation, crop diversification also led to improved nutrient availability in the peanut rhizosphere (Supplementary Data 1). In particular, nitrate ($NO_3^-$-N) and ammonium ($NH_4^+$-N) levels were 52% and 125% higher in crop mixture than in peanut monoculture. This effect was smaller for total N (only 9%), but still significant ($p < 0.05$, Turkey-HSD). In the bulk soil, nutrients (including total N, $NO_3^-$-N, $NH_4^+$-N, total K and available K) did not differ between cropping systems ($p > 0.05$, Turkey-HSD), except organic carbon (SOC), total P and available P which were increased by at least 6%, 12% and 115%, respectively, in crop mixture or rotation compared with peanut monoculture. In general, bulk soil nutrient levels were on average 17–59% lower than the corresponding rhizosphere values ($p < 0.05$), except for total K ($p < 0.001$). The consistently lower soil nutrient availabilities in bulk soil were associated with a significantly lower soil pH in bulk soil, compared to rhizosphere soil (> 0.5 units difference, see Supplementary Data 1, $p < 0.001$, Turkey-HSD). This is typical for the highly weathered red soil at our experimental site, where soil nutrients are easily lost due to the local rapid temperature increases and heavy precipitation (see Methods)[25,26].

### Crop diversification alters the composition of peanut rhizosphere metabolites
Using ultra-performance liquid chromatography-tandem mass spectrometry (UHPLC–MS/MS) we found that of the total 2,891 mass features detected by either positive (50.1%) or negative (49.9%) ionization, 447 metabolic features (15.5% of total) were identified and annotated with >70% fragment score of the m/z Cloud best match using the database (http://www.mzcloud.org)[27,28]. These included benzenoids, hydrocarbon derivatives, lipids and lipid-like molecules, nucleosides and nucleotide analogs, organic acids and their derivates, organic nitrogen compounds, organic oxygen compounds, organohalogen compounds, organoheterocyclic compounds, and phenylpropanoids and polyketides (Fig. S3, Supplementary Data S2)[29]. Principal component analysis (PCA) showed that the annotated metabolites were strongly clustered according to crop diversification, with PC1 and PC2 accounting for 29.7% and 13.5% of the variance, respectively (R = 0.783, $p_{ANOSIM} = 0.001$) (Fig. 2a). A similar grouping emerged from the hierarchical clustering of the Heatmap of metabolic features between treatments (Fig. S3). Using on a twofold change threshold, we found 49 metabolites that differed significantly between crop mixture and peanut monoculture, and 202 metabolites that differed significantly between three-crop mixture and -crop rotation ($p < 0.05$, t-test, Supplementary Data 2). In particular, four metabolites annotated with over 95% fragment score were specifically enriched in crop mixture compared with monoculture and rotation (fold change>2, $p < 0.001$, t-test, Fig. 2b; Supplementary Data 2). Based on primary and secondary mass spectrum analyzes and the comparison of standards' mass spectrum, these four metabolites were putatively identified as quercetin,

hyperoside, scopoletin and syringaldehyde (Fig. S4, S5; Supplementary Data 2). Thus, crop diversification enriched the peanut rhizosphere with specific flavonoids (i.e. quercetin and its 3-o-galactoside hyperoside), coumarins (i.e. scopoletin) and their derivatives (i.e. syringaldehyde).

## Crop diversification enhances peanut root gene expression for the biosynthesis of specific metabolites

To determine the origin of the metabolites that were enriched under crop diversification, we performed root transcriptome sequencing to compare gene expression in peanut roots between the least diverse

(PP) and most diverse system (PM-R). As expected, PCA of transcriptomic data confirmed clearly separated clusters of monoculture and crop mixture (Fig. 2c). In total, 2,911 differentially expressed genes (DEGs) were regulated in the crop mixturegroup compared to the peanut monoculture group, including 1322 downregulated genes and 1,589 upregulated genes (twofold change cut off, q < 0.05, Fig. 2d). Using the Kyoto Encyclopedia of Genes and Genomes (KEGG) database, we found that the pathways of ABC transporters (ko02010), phenylpropanoid biosynthesis (ko00940), cytochrome P450 related metabolism (ko00982 and ko00980) and degradation of other organics or metabolism were ranked among the top 10 KEGG pathways

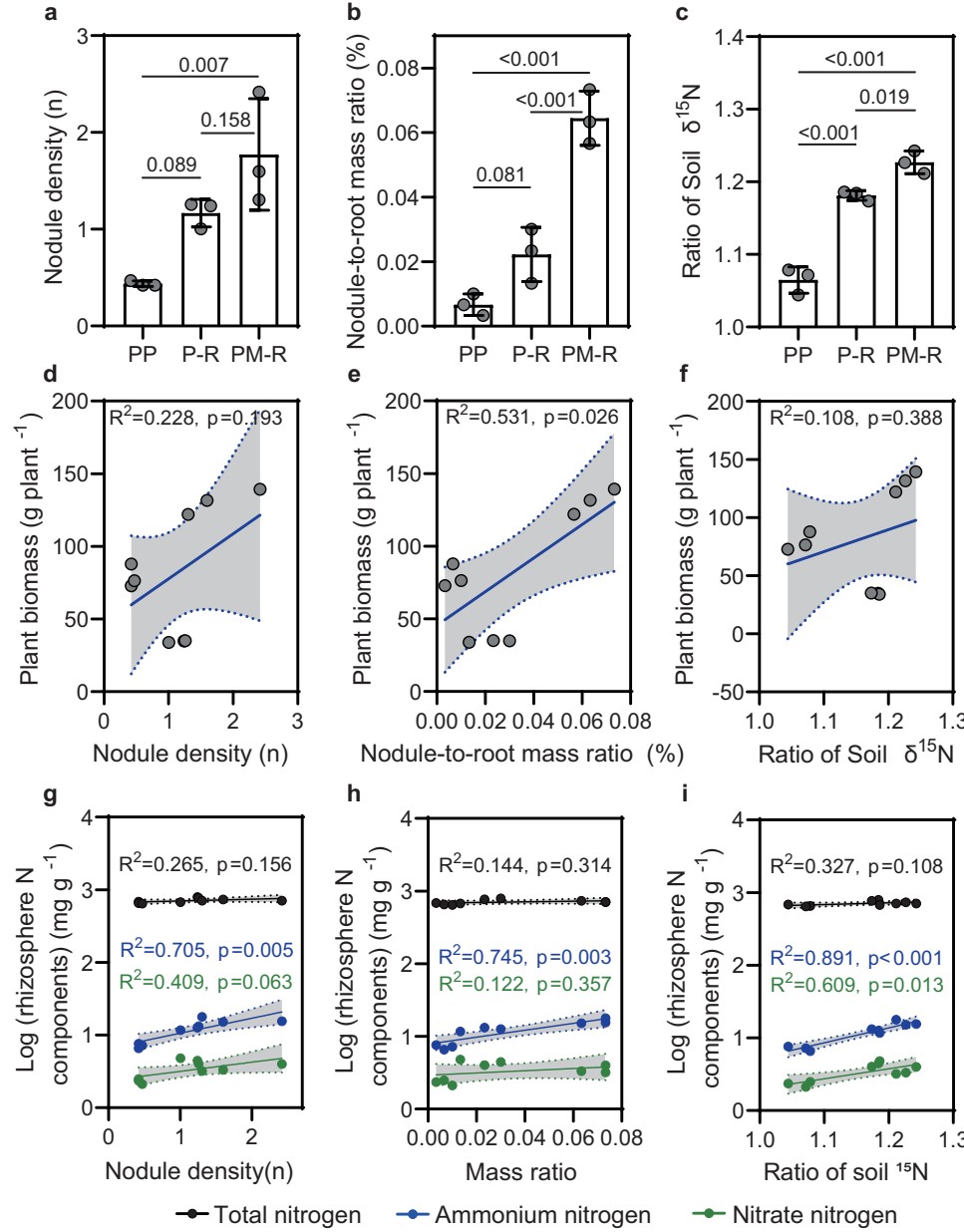

**Fig. 1 | Effect of crop diversification on peanut root nodulation and rhizosphere N availability. a, b** Effect on peanut nodulation. **c** Effect on peanut rhizosphere $^{15}$N fixation. The data in **a**–**c** are shown as the mean ± SD. The error bars with $p$ values between groups were calculated using one-way ANOVA and Tukey's post-hoc tests (two-sided, $n$ = 3 biologically independent replicates from 9 samples per treatment). **d, e** Correlations between root nodulation and peanut plant biomass. **f** Correlations between soil $^{15}$N fixation and plant biomass. **g, h** Correlations between root nodulation and rhizosphere nitrogen components including total (black points with regression), ammonium (blue points with regression) and nitrate (green points with regression) nitrogen. **i** Correlations between soil $^{15}$N fixation and rhizosphere nitrogen components including total, ammonium and nitrate nitrogen. In **d**–**i**, solid lines represent the least squares regression fits and shaded areas with dotted line border represent the 95% confidence intervals. PP, P-R and PM-R represent peanut monocropping, peanut-oilseed rape rotation, and peanut-maize intercropping rotated with oilseed rape, respectively. Source data are provided as a Source Data file.

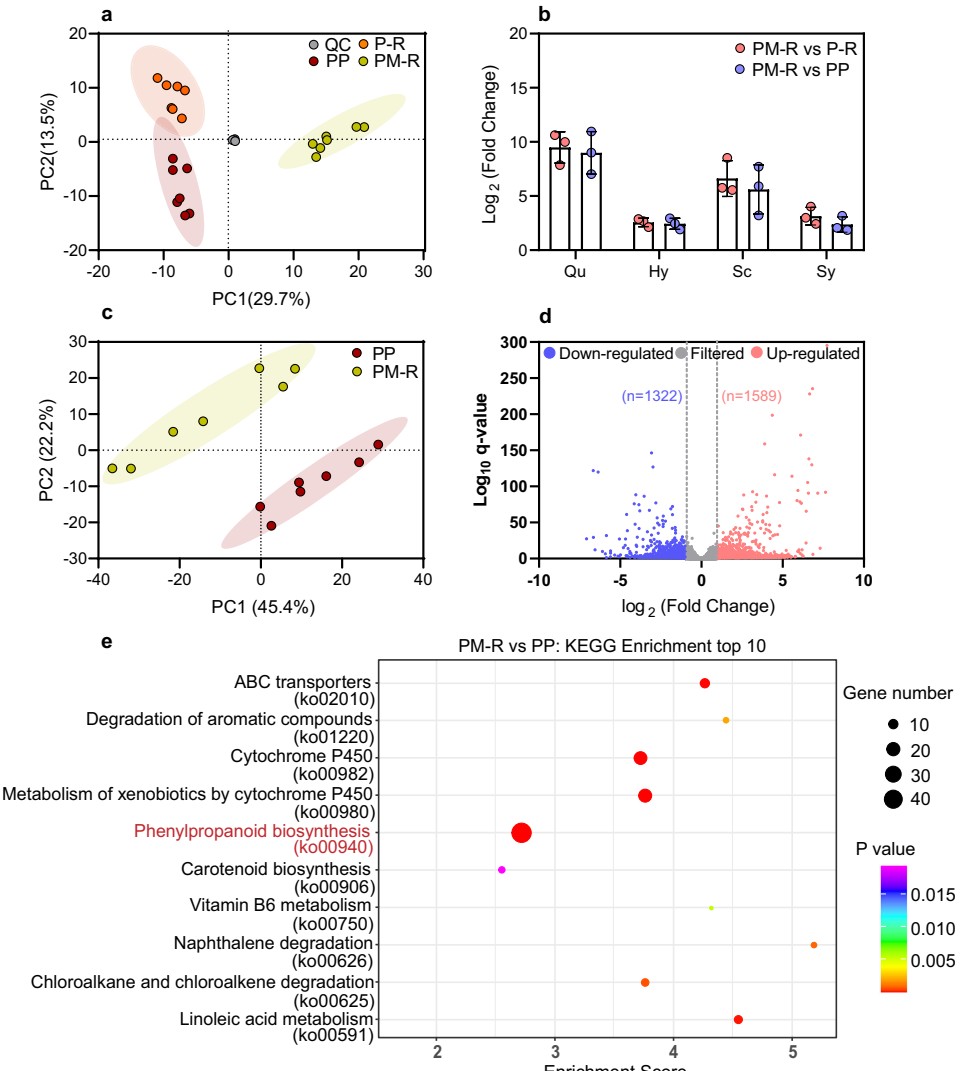

**Fig. 2 | Effect of crop diversification on metabolic production in the peanut rhizosphere. a** Principal component analysis of the metabolites detected in peanut rhizosphere soil. QC, quality control samples (composed of a small aliquot of each sample). **b** Screening for specific enriched metabolites (m/z Cloud database best match >95%). Four metabolites, identified as quercetin (Qu), hyperoside (Hy), scopoletin (Sc), and syringaldehyde (Sy), were selected based on the fold change (>2) of relative concentration by two sided t-test. The data are shown as the mean ± SD ($n = 3$ biologically independent replicates from 7 samples per treatment). **c** Principal component analysis of root transcriptomic variance of PP and

PM-R. Colored shades in **a** and **c** represent 95% confidence ellipses around each group to distinguish community differences ($n = 7$ biologically independent samples per treatment). **d** Volcano plot showing differentially regulated genes in peanut roots in PM-R versus PP. Genes with fold change >2 and q < 0.05 are marked with purple (down-regulated) and pink (up-regulated). **e** Top-10 of enriched Kyoto Encyclopedia of Genes and Genomes (KEGG) pathways of differentially expressed genes (DEGs) in PM-R versus PP. PP, P-R, and PM-R represent peanut monocropping, peanut-oilseed rape rotation, and peanut-maize intercropping rotated with oilseed rape, respectively. Source data are provided as a Source Data file.

that were enriched in crop mixture compared to monoculture ($p < 0.001$, t-test, Fig. 2e). Given the results of the metabolome analyzes (Fig. 2b; Fig. S4), we focused on the phenylpropanoid (ko00940) and flavonoid biosynthesis (ko00941) pathways and found that 48 DEGs involved in the former and eight DEGs involved in the latter were enriched in crop mixture ($p = 2.7 \times 10^{-10}$ and $p = 0.016$, respectively, t-test; Supplementary Data 3 and 4). Based on the functional classification by Gene Ontology (GO), most of these genes in crop mixture were specifically enriched for peroxidase activity (indicating oxidative stress) and flavonoid biosynthetic process (Fig. S6a; Supplementary Data 5). Using quantitative PCR (qPCR) of fifteen representative genes, our results showed that the observed differences in gene expression between monoculture and crop mixture were consistent with the root transcriptome data ($p < 0.05$, t-test, Fig. S6b), except for gene 4GQZ4H (which participates in beta-glucosidase biosynthesis, $p > 0.05$, t-test, Fig. S6b). Collectively, these results provide strong evidences that the

presence of maize and oilseed rape in the crop mixture system induces flavonoid and coumarin biosynthesis pathways in peanut root, resulting in the accumulation of these metabolites in the rhizosphere.

## Crop diversification alters the peanut rhizosphere bacterial community

Given the effects of crop diversification on peanut root metabolic biosynthesis and release (Fig. 2), we next investigated whether these changes in rhizosphere chemistry have the ripple-on effect on the root associated bacterial community. In total, 4649 amplicon sequence variants (ASVs) were detected. Although increasing crop species appears to negatively affected bacterial alpha-diversity, only Shannon index showed significant difference ($p < 0.05$, Turkey-HSD) and no significant differences were found based on the Chao1 index ($p > 0.05$, Fig. 3a, Turkey-HSD). Also, crop diversification resulted in more variability, as shown in the first axis of principal co-ordinates analysis

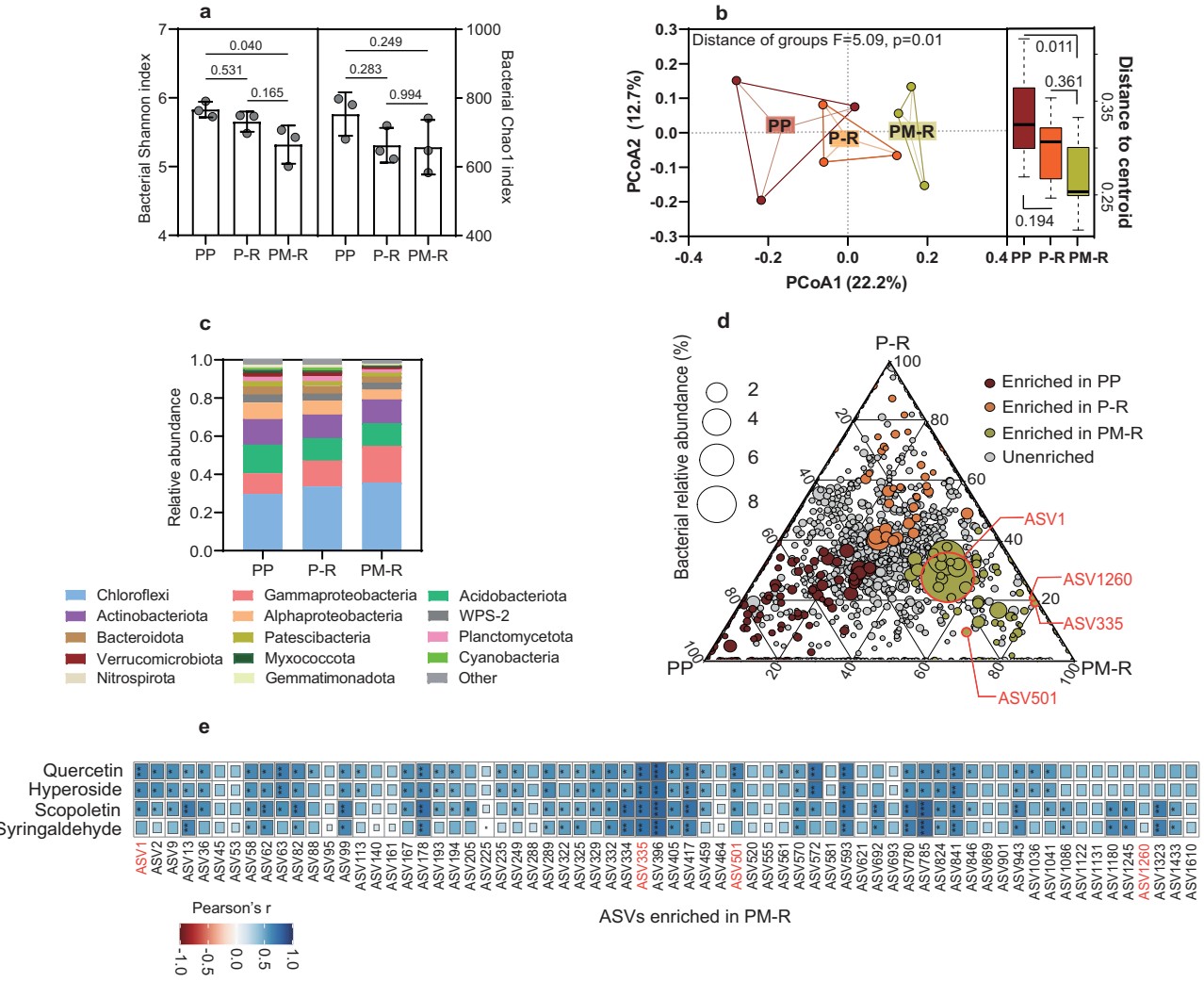

**Fig. 3 | Effect of crop diversification on the peanut rhizosphere bacterial community. a** Shannon and Chao1 richness indices. The data are shown as the mean ± SD. The error bars with *p* values between groups were calculated using one-way ANOVA and Tukey's post-hoc tests (two-sided, *n* = 3 biologically independent replicates from 9 samples per treatment). **b** Principal coordinate analysis (PCoA) of bacterial beta dispersion among different groups based on Bray–Curtis distance (left) and distance of centroid beta-dispersal values for groups (right). Box plots indicate median (black line), 25th, 75th percentile (box), and 5th and 95th percentile (whiskers). *p* values were adjusted using multiple (95% family-wise confidence level) comparisons using Tukey's HSD (*n* = 3 biologically independent replicates from 9 samples per treatment). **c** Phylum-level distribution of ASVs. **d** Ternary plot of bacterial ASVs shared among the different peanut rhizosphere communities. Circle sizes represent the relative abundances of the bacterial ASVs identified. Circles with red borders represent ASVs with the same marker sequence as the subsequent bacterial isolates (Fig. 4a). **e** Heatmap of specific enriched metabolites and PM-R enriched bacterial ASVs according to Pearson's correlations. Positive and negative correlations are shown in blue and red, respectively. \**p* < 0.05, \*\**p* < 0.01, \*\*\**p* < 0.001. PP, P-R, and PM-R represent peanut monocropping, peanut-oilseed rape rotation, and peanut-maize intercropping rotated with oilseed rape, respectively. Source data are provided as a Source Data file.

of multivariate homogeneity of group dispersions (Fig.3b, *p* = 0.01, Turkey-HSD). Relative to the monoculture, microbiome composition heterogeneity in peanut rhizosphere decreased with crop diversification (*p* = 0.011-0.361, Turkey-HSD), indicating that different crop co-existence induces peanuts to make root-associated microbial composition more homogenous. At the level of individual taxa, crop diversification was found to increase the relative abundances of some dominant phyla (average relative abundance >5%) such as Chloroflexi and Gammaproteobacteria, while Alphaproteobacteria decreased (Fig. 3c and Supplementary Data 6, *p* < 0.05, Turkey-HSD). In particular, several ASVs belonging to Gammaproteobacteria were enriched in the peanut rhizosphere in crop mixture group (Fig. 3d; Fig. S7; Supplementary Data 7). Linear correlation analysis showed that the relative abundances of three-quarters (49 of 68) of crop mixture group enriched ASVs were positively correlated with at least one of specific flavonoids, coumarins and derivatives (Fig. 3e).

## Crop diversity-enhanced rhizosphere metabolites trigger bacterial nitrogen fixation

As we expected, the rhizosphere bacterial community was selectively influenced by the production and exudation of active specific metabolites from peanuts in the system with maize and rape. However, it is not clear why peanut alters its rhizosphere chemistry and microbiota. It could be related to the higher plant nitrogen fixation exhibited in the field of crop mixture group. To test this hypothesis, we isolated and cultivated 109 bacterial species from the peanut rhizosphere in crop mixture and tested the responses of selected isolates, including free-living and symbiotic N₂ fixers, to specific metabolites in microplate incubation assays (Figs. 4, 5). These isolates represented five phyla: Firmicutes (54.1%), Actinobacteria (19.3%), Gammaproteobacteria (12.8%), Alphaproteobacteria (6.4%) and Betaproteobacteria (5.5%) (Fig. 4a). From this set we selected and purified 27 isolates for microplate incubation assays (Supplementary Data 8), which included four

strains (strains N4, OP14, N68 and N69) affiliated with Pseudomonadales and Burkholderiales that had similar marker sequences as the ASVs enriched in crop mixture group (ASV1, ASV335, ASV501, ASV1260 marked in Fig. 3d) (Supplementary Data 9). Of these four representative strains, three (N4, N68 and N69) were able to grow on a nitrogen-free medium, indicating their capacity for free-living $N_2$ fixation. In addition, we purified four isolates belonging to *Rhizobium* and *Bradyrhizobium* (N43, N45, N47 and N59). Although these strains were not phylogenetically consistent with specifically enriched ASVs in crop mixture group, their potential for host invasion and colonization and presence in the peanut rhizosphere could be related to the increased host nodulation and rhizosphere N availability observed in crop mixture group (Fig. 1). Finally, we selected 19 isolates of other species, whose marker sequences were not similar to the enriched ASVs, to serve as controls.

Using these 27 strains, we conducted microplate incubation assays to evaluate whether bacterial growth rates were influenced by the metabolites that were specifically enriched in the rhizosphere of peanut in mixture: quercetin, hyperoside, scopoletin and syringaldehyde (Fig. S8). On average, these individual metabolites were found to affect the growth rate (V), either positively or negatively, of 48% of the tested strains in 1/5 TSB medium containing 5 µg mL$^{-1}$ of one of different metabolites (Fig. 4b–e, $p < 0.05$, t-test). Interestingly, none of the metabolites had a negative effect on the growth rates of the four representative isolates from Pseudomonadales and Burkholderiales. Among these strains only neutral or positive effects were detected (N4, $p = 0.03–0.50$; N68, $p = 0.0003–0.04$; N69, $p = 0.01–0.23$; OP14, $p = 0.02–0.66$, t-test). In addition, for the free-living $N_2$ fixers in this group (N4, N68 and N69), their $N_2$-fixing capacities under metabolite addition were consistently positively correlated with their growth rates (Fig. 4f–h).

In contrast to the generally positive responses of free-living $N_2$ fixers to the tested metabolites, the growth rates of the four rhizobial isolates (i.e. potential symbiotic $N_2$ fixers) were generally insensitive or even showed a negative response to flavonoids and coumarins (N43, $p = 0.0001–0.64$; N45, $p = 0.04–1.00$; N47,0.30–0.84; and N59, $p = 0.01–0.95$, t-test), except for N43 which responded positively to scopoletin ($p = 0.012$, t-test) and syringaldehyde ($p < 0.001$, t-test). Since flavonoids are common signals for the establishment of legume-rhizobia symbioses[6], we wondered about the poor growth response of rhizobia to these metabolites. Considering the large increase in peanut nodulation in crop mixture (Fig.1), we hypothesized that increased rhizodeposition of flavonoids and their derivates in three-crop mixture enhanced the rhizobia's ability to colonize host roots, rather than their growth rates. To test this, we first inoculated peanut seedlings with the individual $N_2$-fixing rhizobial isolates (N43, N45, N47 and N59), and found that only *Bradyrhizobium* N47, through lateral root base invasion and intra- and inter-cellular colonization[30], successfully induced nodulation of peanut host plants within 30 days (Fig. 5a, Fig. S9a–d). Therefore, we focused on this strain to assess whether the enriched root metabolites increased nodulation signaling during the establishment of the symbiosis. Indeed, the addition of flavonoids and coumarin to pure *Bradyrhizobium* cultures increased the expression of Bradyrhizobial *nodD1* and *nodC* genes by 21-126% (*NodD1*) and 216-430% (*NodC*), compared to controls (Fig. 5b, c). Similar stimulating effects on the nod genes (*NodD1* and *NodC*) have also been observed in the model microorganism *Sinorhizobium meliloti* (strain1021) for such metabolites (Fig. S10). Simultaneously, the addition of these metabolites to peanut seedlings inoculated with *Bradyrhizobium* N47 enhanced peanut root gene expression of *AhSYMRK*, *AhCCaMK* and *AhNIN* by 43-169% at the transcriptional level compared to *Bradyrhizobium* inoculation without added metabolites (Fig. 5d–f). These upregulated plant genes play crucial roles in nodule organogenesis by encoding leucine-rich repeat receptor-like kinase (*SYMRK*), calcium spikes by a calcium calmodulin-dependent protein kinase (*CCaMK*)

and an RWP-RK transcription factor (*NIN*)[31,32]. Together, the metabolite-triggered increases in bacterial and root gene expression eventually resulted in a 21-90% increase in nodule number, measured 30 days after inoculation and metabolite addition (Fig. 5g). Thus, it appeared that the flavonoids and coumarins acted as a chemical communication signal between peanut and *Bradyrhizobium*. The fourth metabolite, syringaldehyde, did not have such effect (Fig. 5b–g, $p > 0.05$, Turkey-HSD). Overall, these results suggest that peanut rhizosphere metabolites produced under the influence of coexisting crops enhance the peanut- *Bradyrhizobium* symbiosis, but not growth rate.

## Discussion

In both natural and agricultural ecosystems, belowground facilitation between legume and non-legume plants has been found to regenerate soil fertility, especially N availability[33–35]. Many studies simply attribute this positive plant-soil feedback to the innate function of legumes to fix atmospheric $N_2$ and focus on how this service improves the productivity of non-legume plants, such as maize and wheat in crop systems (e.g., Li et al and Zhao et al)[23,36]. Here, we shift the focus to the legumes themselves to examine how their capacity to contribute N to the system is influenced by coexisting crops. Data from our field study showed that peanut biomass, root nodulation (including nodule density and nodule-to-root mass ratio) and soil $^{15}N_2$ fixation were significantly increased in the most diverse system (including both rotation with oilseed rape and intercropping with maize), compared to the peanut monoculture and peanut-oilseed rape rotation without maize intercropping (Fig.1: PM-R versus PP and P-R). Moreover, the increased nodulation and free-living $N_2$ fixation were positively correlated with rhizosphere nitrogen accumulation. These findings suggested an interspecific positive feedback, from maize in particular, on peanut $N_2$ fixation.

To understand the mechanisms underlying the increased nodulation, peanut N uptake and N availability, we zoomed in on the peanut roots and rhizosphere. Chemical analysis of peanut rhizosphere soil showed that peanut grown in the most diverse cropping system accumulated specific metabolites in its rhizosphere: flavonoids (quercetin and hyperoside), coumarins (scopoletin) and their derivatives (syringaldehyde). These active secondary metabolites, which many plant species are able to produce, are synthesized via the phenylpropanoid pathway[37,38] and regulated by cytochrome P450 enzymes and ATP-binding cassettes (ABC)-transporters for flavonoids and coumarins biosynthesis and transport[38,39]. Coincidentally, these pathways (KEGG pathway top-10) were significantly enriched in peanuts grown in the most diversified system. By examining the increased relative expression of functional genes that participate in the specific metabolic biosynthesis and transport processes in peanut roots (Fig. S6b), it appeared that the metabolites were mainly produced by the peanut plants their own. Although we cannot completely exclude the possibility that some of the measured metabolites were originated from gradient diffusion from neighboring (maize) or previously grown (rape) plants, soil pore space structure and soil microbial substrate consumption have been reported to greatly diminish the diffusion efficiency of these soluble compounds from neighboring or historical plant species at long distance (such as >10 cm) in soil[9,40,41].

A relevant question is why the peanut rhizosphere accumulated these metabolites specifically when peanut was intercropped with maize. Shaped through a long evolutionary process, rhizosphere deposition is among a plant's most sophisticated strategies to adapt to changing environments[14]. Flavonoids and coumarins, the secondary metabolite deposits that we found to be increased in the rhizosphere of peanuts co-grown with maize, are known to be produced by plants in response to environmental changes, including light intensity, ultraviolet radiation, temperature variation and drought[42]. Shade from the maize canopy and associated competition for light may have

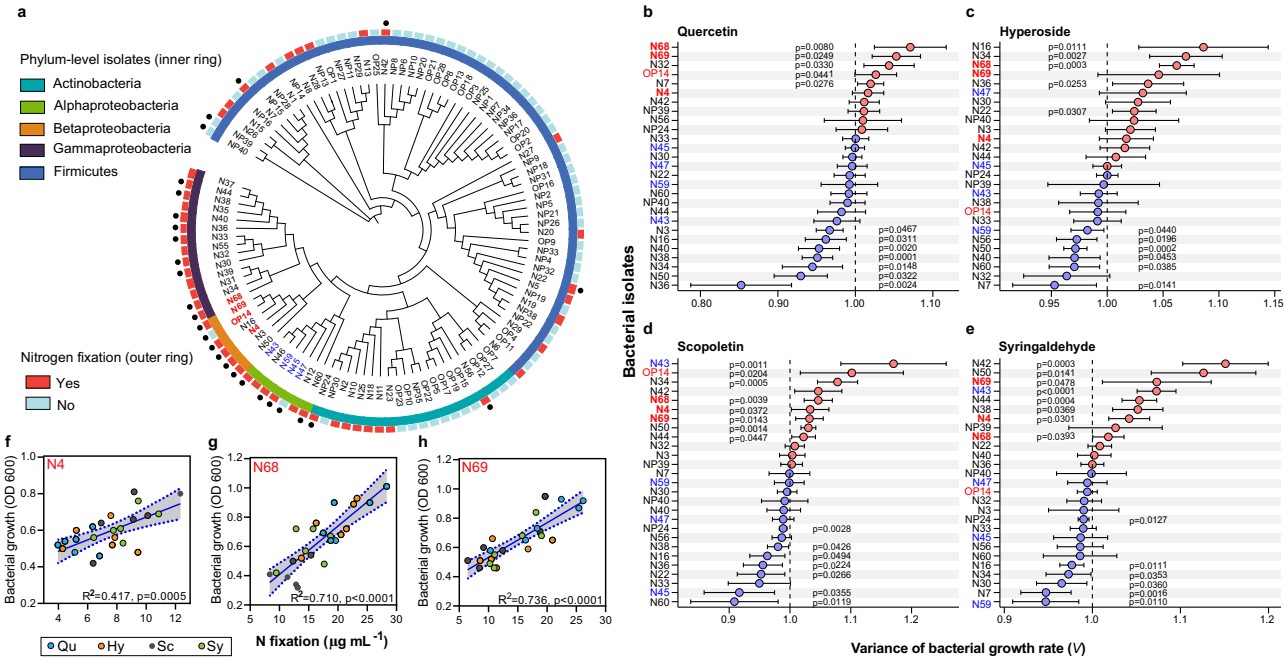

**Fig. 4 | Bacterial isolates from the PM-R peanut rhizosphere: effects of typical metabolites on bacterial growth rates, free-living N₂ fixation. a** Cladogram showing the phylogenetic relationships among 109 heterotrophic bacterial isolates from the PM-R rhizosphere and their potential for N₂ fixation. The leaf labels indicate the representative sequence IDs, with red labels indicating the four selected isolates that were phylogenetically consistent with PM-R-enriched ASVs at the order and genus levels, and blue labels indicating the four selected rhizobia isolates. The rings, from the inner to outside circles, represent (1) the phylum level taxonomy of isolates; (2) the capacity for bacterial nitrogen fixation; and (3) the strains selected for microplate incubation with typical rhizosphere metabolites.

**b–e** Effect of rhizosphere metabolites on the growth rate (V) of selected bacterial strains, measured in microplate assays during the bacterial logarithmic growth phase ($n = 6$ biologically independent samples). V > 1 represents growth promotion with metabolite addition; V < 1 represents growth inhibition. Bars with $p$ values represent significant differences as determined by two-sided t-test.
**f–h** Correlations between bacterial growth of free-living N₂ fixers and their capability for N₂ fixation. Solid lines in f-h represent the least squares regression fits and shaded areas with dotted line border represent the 95% confidence intervals. Qu quercetin, Hy hyperoside, Sc scopoletin, Sy syringaldehyde. Source data are provided as a Source Data file.

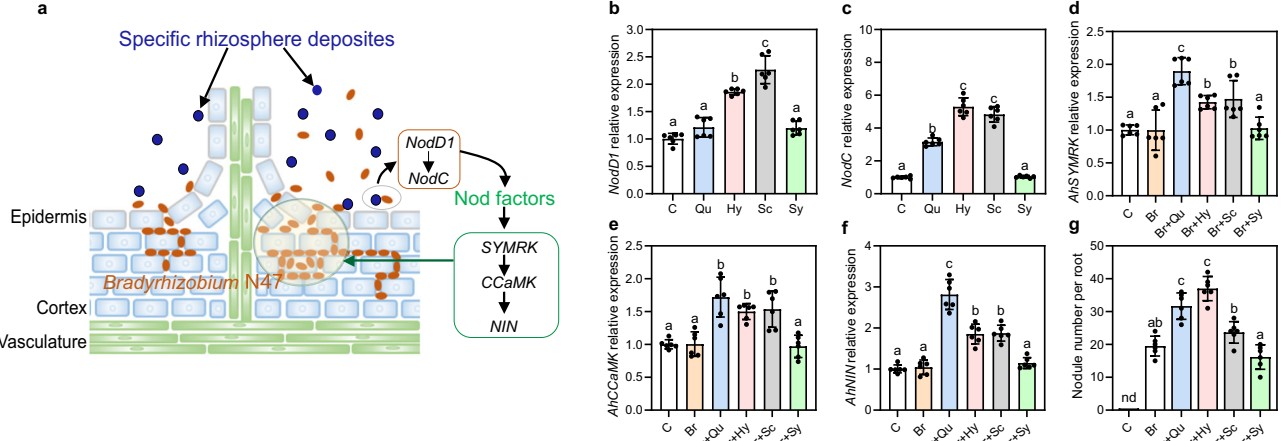

**Fig. 5 | Bacterial isolates from the PM-R peanut rhizosphere: effects of typical metabolites on *Bradyrhizobium* N47 colonization. a** Diagram of symbiosis signaling pathway of *Bradyrhizobium* N47 in peanut root, inducing nodule formation at lateral root bases. **b, c** Expression of bradyrhizobial nodulation signaling genes, 48 h after addition of metabolites. **d–f** Expression of host common symbiotic signaling genes, measured in peanut roots 24 h after bacterial inoculation and metabolite addition. **g** Peanut root nodule number 30 days after bacterial inoculation

and metabolite addition. The data in **b–g** are shown as the mean ± SD. The error bars with lowercase represent significant differences between groups ($p < 0.05$) via one-way ANOVA and Tukey's post-hoc tests (two-sided, $n = 6$ biologically independent samples). For exact statistical values, see Supplementary data 12. C, control; Br *Bradyrhizobium* N47, Qu quercetin, Hy hyperoside, Sc scopoletin, Sy syringaldehyde. Source data are provided as a Source Data file.

triggered the peanut transcriptional response and induced flavonoid production[43,44]. In addition, maize root exudates may have enhanced peanut flavonoid biosynthesis, as suggested by Li et al in a faba bean-maize intercropping system[21]. Unlike the neighboring maize effect, peanuts that were rotated with oilseed rape did not result in biomass promotion, despite increasing nitrogen fixation in soil. This suggests that, in addition to resource availability, other factors in the legacy effect stemming from historical species may play a role in determining peanut development[18,19,22].

Apart from their origin, we also shed light on the function of these metabolites, i.e. their role in the chemical dialog between peanut and its rhizosphere microbial community. In general, we found that the higher specific metabolic deposition in the peanut rhizosphere triggered by crop diversity had a greater impact on bacterial beta diversity than alpha diversity. Taxonomic analysis showed that ASVs affiliated with the orders Pseudomonadales and Burkholderiales were specifically enriched in the peanut rhizosphere of the most diverse system, PM-R. Three of the isolated strains (N4, N68, and N69), with marker sequences similar to the enriched Pseudomonadales and Burkholderiales ASVs in PM-R, were found to be capable of free-living $N_2$ fixation. Remarkably, their growth and correlated level of $N_2$-fixation were promoted by the specific metabolites that had accumulated in the peanut rhizosphere of the most diverse system. Thus, the positive growth response of these strains to plant diversity-driven specific metabolites may have synergistically promoted free-living $N_2$ fixation in the peanut rhizosphere, as reflected by the increased rhizosphere N availability observed in the field. In the case of these $N_2$-fixers, specific flavonoids and coumarin act more like as carbon resources, supporting microbial survival and nitrogen fixation. Given the potentially antimicrobial effect of flavonoids[4,38], the positive response of the observed strains to flavonoids and coumarins suggested their adaptive plasticity under the filtering effect of rhizosphere chemical selection[45].

The observed chemical interactions between peanut and rhizosphere bacteria involved not only free-living but also symbiotic $N_2$-fixers. Focusing on *Bradyrhizobium* (N47, which was isolated from the peanut rhizosphere and tested for symbiotic potential), we found that the metabolites (quercetin, hyperoside and scopoletin) helped to initiate the plant-microbe symbiosis and thus aided the survival of both by nodulation. This is in line with findings that flavonoids can act as a chemical language between rhizobia and legumes to initiate root nodulation[6,46]. In this dialog, the microbial *Nod* genes regulate the production of Nod factors (lipochitooligosaccharide)[6,31], which in turn activate plant downstream signalling genes involving *SYMRK, CCaMK* and *NIN*. These factors are needed to trigger the nodule developmental program[31,32] which in peanut initiates at the lateral root base (Fig. S9c)[30,31]. Thus, the specific deposits that accumulated in the peanut rhizosphere of the most diverse cropping system helped to activate a common symbiosis signaling pathway. This molecular-level finding may explain the increased nodule density with crop diversification observed in the field (Fig. 1).

It is worth noting that scopoletin did induce the expression of bradyrhizobial genes involved in root nodule-forming symbioses but did not increase the number of root nodules compared to the control treatment with bradyrhizobial inoculation alone (Fig. 5). To the best of our knowledge, evidence for the role of coumarins in root nodulation or $N_2$-fixation is scarce[47]. When we used the model plant *Medicago truncatula* (A17) to establish symbiosis with its rhizobia *Sinorhizobium meliloti* (strain1021), the addition of trace scopoletin (SML, 5 µg mL$^{-1}$) resulted in the upregulation of genes involved in root nodulation (including *MtCCaMK, MtNIN, MtERN1, MtVAPYRIN, MtENOD11, MtRIP1* and *MtFLOT4*) and plant defense (including *MtPR4, MtPR10* and *MtGST*) ($p < 0.05$). This ultimately led to an increase in both root nodules and biomass in Madicago compared with control (C) and single bacterial inoculation (SM). Comparatively, the addition of high scopoletin (SMH, 50 µg mL$^{-1}$) weakens the effect on plant nodulation and defense

compared with low scopoletin addition (Fig. S11). Some coumarins (such as scopoletin and esculetin) are recognized for their efficiencies as antibiofilm and antimicrobial compounds, contributing to plant immunity[38]. This rhizosphere selective effect through specific metabolites could promote microbially elicited plant-microbe interactions, providing benefits for nutrient acquisition in the rhizosphere and plant health[47].

Interestingly, whether flavonoids and coumarins induced by maize and oilseed rape coexistence or gaseous ethylene which was previously found to be stimulated by non-legume species recognition[17], these plant secondary metabolites have been reported to not only involved in the plant defense of biotic and abiotic stresses but also act as the chemical cue for "belowground cry-for-help": specific components of root exudates to recruit root-associated and soil-associated microbiomes to enhance plant fitness[14,48]. Here, we provide a mechanistic understanding of why specific crop combinations may show higher levels of leguminous $N_2$ fixation than expected. By combining a field investigation of multi-omics (rhizosphere metabolome analysis and peanut root amplicon sequencing) with a molecular-level analysis of laboratory plant-microbe systems, we showed that crop diversification, specifically intercropping with maize, induces a chemical dialog in which peanut plants select specific free-living and symbiotic $N_2$-fixing bacteria through rhizosphere metabolite deposition (Fig. 6). The substrate adaptation of free-living $N_2$-fixers and changes in symbiont life strategies in response to such chemical cues ultimately determines rhizosphere functions in the holobiont comprised of the host plant and its microbiota[49]. Such plant-microbe functional alliances, with their mutual fitness benefits, present a new perspective for understanding the relationship between aboveground plant diversity and belowground ecosystem functioning[33,45,50]. Finally, our findings also provide relevant mechanistic insights for the design of intelligent crop combinations and targeted manipulation of the rhizosphere microbiome and functionality in sustainable agroecosystems.

## Methods
### Field experiment site and design
For this study, samples were collected from a long-term field experiment, established in 2011 at the Yingtan Red Soil Ecological Experimental Station of the Chinese Academy of Sciences in Jiangxi Province, China (28°12′N, 116°55′E). The climate is subtropical (see Chen et al.[17] for more detailed description). The soil is an acidic, loamy clay derived from Quaternary Red Clay and classified as a Ferralic Cambisol[51]. Before the establishment of this experiment, the site was a mass pine (*Pinus massonina* Lamb.) natural secondary forest.

The field experiment compared three crop systems based on common practices in local intensive agricultural systems of tropics and subtropics[36,52]: (I) a single-crop system (PP, monoculture of peanut, *Arachis hypogaea*), (II) a two-crop system (P-R, rotation of peanut and oilseed rape *Brassica campestris*), and (III) a three-crop system (PM-R, intercropping peanut and maize *Zea mays* L., rotated with oilseed rape) (Fig. S1). Each treatment was carried out in three randomized blocks, with three replicate plots of 20 m × 5 m × 1.5 m (width × length × depth), each plot further divided into three subplots. The plots were separated by 10 cm (thickness) concrete baffle plates, and the subplots by ridges.

### Field cropping practices
The design of the three-crop system (PM-R, intercropping peanut and maize, rotated with oilseed rape) during the spring-summer season (April-August) included a 1.0 m peanut strip (2 rows of peanut, with a 0.5 m interrow distance) and a 1.0 m maize strip (2 rows of maize, with a 0.5 m interrow distance). The interplant distance within the same row was 0.2 m for peanuts and 0.25 m for maize. In the single-crop system (PP, monoculture of peanut) and two-crop system (P-R, rotation of

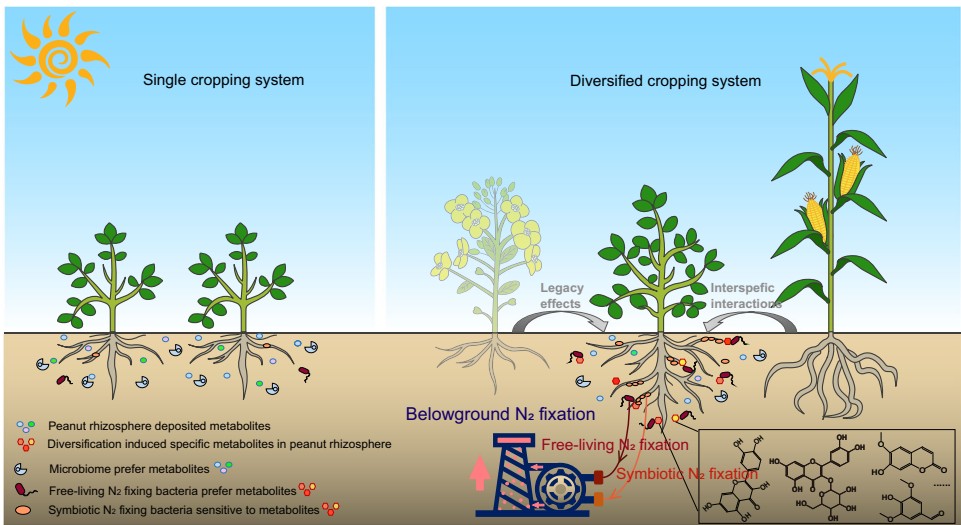

**Fig. 6 | Effect of crop diversification on metabolite deposition and biological N$_2$ fixation in the leguminous rhizosphere.** The precrops (e.g. legacy effects from oilseed rape) and neighbors (e.g. interspecific interactions of maize) induce changes in the metabolites deposited into the legume rhizosphere. The deposited flavonoids and coumarins selectively influence the functional activities of free-living and symbiotic N$_2$ fixers which enhance rhizosphere N$_2$ fixation in the field.

peanut and oilseed rape) treatments, the interrow and interplant distances for peanut were 0.5 m and 0.2 m, respectively, which made the peanut density identical to that in a comparable area of the PM-R treatment. During the autumn-winter season (September-March), the interrow and interplant distances for oilseed rape were 0.5 m and 0.2 m, respectively, in the P-R and PM-R treatments. Peanut and maize were sown on the 1st–15th of April, and oilseed rape was sown on the 1st–15th of September. The topsoil (0–25 cm) was ploughed before cultivation every year. All plots were irrigated and weeded during the growing period.

## Field soil characteristics and annual fertilizations

The basic soil characteristics of the field experiment in 2012 were as follows: organic matter 4.58 g kg$^{-1}$, total N 0.45 g kg$^{-1}$, total P 0.35 g kg$^{-1}$, total K 11.84 g kg$^{-1}$, available P 1.68 mg kg$^{-1}$, available K 54.17 mg kg$^{-1}$, NH$_4^+$-N 5.24 mg kg$^{-1}$, NO$_3^-$-N 2.59 mg kg$^{-1}$, and pH 4.84. All treatment groups received urea (containing 46% N), calcium superphosphate (containing 12.5% P$_2$O$_5$) and potassium chloride (containing 60.0% K$_2$O) chemical fertilizers at rates of 150 kg ha$^{-1}$ y$^{-1}$, 75 kg ha$^{-1}$ y$^{-1}$ and 60 kg ha$^{-1}$ y$^{-1}$, respectively, 10-15 days before seeds were sown in the spring. In the oilseed rape planting season (P-R and PM-R treatments), 1/2 doses of chemical fertilizers were applied before seed sowing.

## Collection of soil and plant samples

Soil and peanut plant samples were collected on 10 June 2019 at the peanut flowering stage. To sample rhizosphere soil, we randomly selected six peanut plants from each subplot and gently brushed off the soil adhering to the root systems, resulting in one composite rhizosphere sample per subplot ($n = 9$ per treatment). The removed plants were used for determining plant height, biomass and nodulation. To sample the bulk soil, we collected six soil cores in each subplot (5–20 cm depth), 25 cm away from crop roots, using an "S" sampling pattern, and then pooled the cores into one composite bulk soil sample per subplot ($n = 9$ per treatment). All samples were immediately sieved at 4 mm to remove plant debris and stones. From each soil sample, 3 g was stored at –80 °C for microbial molecular analysis, while another 20 g was stored at 4 °C for soil chemical analyzes. The soil chemical properties were characterized by standard methods below. In addition, 3 g of each rhizosphere soil sample was frozen in liquid nitrogen and transported on dry ice for soil metabolic analysis. Of the latter group, two samples per treatment were set aside for pre-analysis but were subsequently lost due to a broken freezer; hence for the metabolite analysis $n = 7$ instead of $n = 9$. Finally, 5 g of each rhizosphere sample from the PM-R treatment was stored at 4 °C for functional bacterial isolation.

Based on the results of the rhizosphere metabolic analysis, we collected fresh peanut root samples from the PP and PM-R treatments on 15 June 2020 for root transcriptome analysis and gene expression. For each treatment, seven randomly selected peanut plants (2–3 plants per replicate plot) were dug out using a shovel. Roots were immediately separated from the plants and washed twice with sterile water. Next, tissues 3–9 cm from the root bases were cut into small pieces (2 cm length), frozen in liquid nitrogen, and transported to the laboratory for RNA extraction, transcriptome sequencing and quantitative reverse transcription polymerase chain reaction (qRT-PCR).

## Soil chemical properties measurement

The soil chemical properties were measured as follows: the pH was determined with a glass electrode and a water-to-soil ratio of 2.5:1 (v:w). The SOC content was determined by the Walkley-Black wet digestion method[53]. The TN and nitrate and ammonium nitrogen (NO$_3^-$-N and NH$_4^+$-N, respectively) concentrations were measured by the Kjeldahl method[54]. Total phosphorus (TP) was digested with HF-HClO$_4$, AP was extracted with sodium carbonate and sodium bicarbonate and then determined with the molybdenum blue method[55], and AK was determined by flame photometry after extraction with ammonium acetate[56].

## Determination of plant physiological indexes

Fresh peanut plants were immediately transferred to the laboratory to measure plant height and biomass after removing soil attached to the roots. For the quality and quantity of nodule density, we measured the number of nodules (n), the biomass of nodules and root per plant, and the cumulative length of each plant root (diameter > 1 mm).

$$\text{Nodule density (n 10 cm}^{-1}) = \frac{\text{The number of nodules per plant}}{\text{Cumulative root length of each plant}} \times 10 \quad (1)$$

$$\text{Nodule to root mass ratio} = \frac{\text{The nodule biomass per plant}}{\text{The root biomass per plant}} \quad (2)$$

## Soil incubation for $^{15}$N detection

Peanut rhizosphere soil (10 g) from PP, P-R and PM-R field treatments was spread flat in small glass serum vials (50 mL). Each vial was sprayed with 1 mL distilled water and then underwent a pre-incubation period (25 °C for one day) to activate soil microbiota. At the start of the incubations, an anaerobic indicator strip was placed in the vial to monitor $O_2$ depletion before closing the vial with butylrubber stoppers and aluminum crimp seals. Then, 20 mL of the gas phase in each vial was evacuated and replaced with an artificial gas mix containing 78% $^{15}N_2$ (99% atom) and 22% $O_2$ using a syringe. Vials were placed in the dark at 28 °C for 7 days. On the fourth day of cultivation, vials were opened for natural gas exchange and then sealed again. The same volume of gas (20 mL) was replaced with an artificial gas mix again. After incubation, soil was collected and sieved at 0.15 mm for soil $\delta^{15}N$ analysis. Each soil treatment was conducted with nine replicates. Soil of PP treatment that was incubated with natural air instead of the artificial gas mix was processed identically and detected as the control.

## Soil $^{15}$N detection

Soil $\delta^{15}N$ analysis was carried out using an Isolink NC elemental analyzer (EA; Thermo Scientific, MA, USA) coupled under continuous flow ConFlo IV (universal continuous flow interface) to the DELTA V Advantage Isotope Ratio Mass Spectrometers (IRMS) (IRMS; Thermo Scientific, MA, USA). Soil samples (ca. 30 mg) were weighed into tin capsules for sample combustion and subsequent reduction over heated copper (Cu) wires within the EA[57]. The resulting $N_2$ was transferred into the IRMS to determine $\delta^{15}N$ values, where isotope ratios were calculated as $\delta^{15}N$.

The delta ($\delta$) notation is conventionally used to express the difference in isotope ratios of the sample (sa), relative to a reference standard (st)-- $R_{sa}$ and $R_{st}$, respectively.

$$\delta^{15}N_{(‰)} = \frac{R_{sa} - R_{st}}{R_{st}} \times 1000 \qquad (3)$$

where the isotope ratio, R is the amount of heavy isotope over the amount of lighter isotope.

## Bacterial high-throughput sequencing

Bacterial community composition was assessed using 16 S rRNA high-throughput sequencing. DNA was extracted from 1.0 g of soil using the FastDNA SPIN Kit according to the manufacturer's instructions. The quality and quantity were measured using a NanoDrop spectrophotometer (NanoDrop Technologies, Wilmington, DE, USA). The V3-V4 regions of the bacterial 16 S rRNA gene were targeted with the primer pair 338 F (5′-ACTCCTACGGGAGGCAGCAG-3′) and 806 R (5′-GGACTACHVGGGT WTCTAAT-3′)[58]. The PCR systems and conditions were consistent with Duan et al. [58]. Amplicon sequencing libraries were constructed using the MiSeq Reagent Kit v3 according to the manufacturer's instructions. High-throughput paired-end sequencing was performed on the Illumina HiSeq 2500 platform (Illumina, San Diego, CA, USA) by OE Biotech Co., Ltd. (Shanghai, China).

The adapt and primer sequences were removed using Cutadapt (v4.4)[59]. Then DADA2 (Divisive Amplicon Denoising Algorithm 2) (version 1.26) was used to merge, filter, trim, and denoise the raw data, and finally generate amplicon sequence variants (ASVs)[60]. Taxonomic assignment for the ASVs was performed using RDP Classifier against the SILVA rRNA database (version 138)[61]. After removing mitochondria and chloroplast, samples were rarefied to the same sequencing depth (16000) for further analysis.

## Rhizosphere soil nontargeted metabolic analysis

To determine the effect of crop diversification on metabolic deposition in the peanut rhizosphere, the metabolic composition of peanut rhizosphere soil samples was determined by nontargeted metabolic profiling, using ultra-performance liquid chromatography-tandem mass spectrometry (UHPLC–MS/MS). Each sample was ultrasonically extracted in an acetonitrile diluent before the analysis. A nontargeted approach with a Q-Exactive quadrupole-Orbitrap mass spectrometer was used to identify metabolites[62]. Mass spectra were acquired in positive and negative ionization modes through full MS and higher energy collisional dissociation (HCD) data-dependent MS/MS analysis (full MS-ddMS2). The mass ranged from 50-2000 m/z. Data were acquired using Xcalibur 2.1 software (Thermo Scientific, Rockford, USA). The datasets from the Q-Exactive analysis were processed with a metabolomics processing workflow using Compound Discoverer 3.0 software (Thermo Scientific, San Jose, CA, USA) to match the primary and secondary mass spectra and retention time (RT) from the database. The advanced Mass Spectral Database (mzCloud, https://www.mzcloud.org/), HMDB and ChemSpider (http://www.chemspider.com/) were chosen as references for the nontargeted metabolomics workflow. All calibration processes and further data analysis were performed using R 4.0 software[63].

## Details of the nontargeted metabolic analysis and standard validation

Freeze-dried soil samples (1 g for each) were vortexed at maximum speed in 10 mL of acetonitrile: isopropanol: water (volume ratio 1:1:1) for 1 min and ultrasonic oscillation for 30 min at 4 °C. The mixture was centrifuged at 13000 x $g$ at 4 °C for 15 min. The supernatant was then held at -20 °C for 1 h for protein precipitation. The suspension was centrifuged again. The supernatant was vacuum dried at 4 °C and re-dissolved in 200 μL 50% acetonitrile (v:v = 1:1 acetonitrile: water). The solution was vortexed and centrifuged at 13000 x $g$ at 4 °C for 15 min. The supernatant was filtered through a polytetrafluoroethylene filter (0.2 μm) for further ultra-high-performance liquid chromatography-tandem mass spectrometry (UHPLC-MS/MS) analysis. Liquid chromatography analysis was performed on a Thermo Dionex Ultimate 3000 DGLC (Thermo Scientific, Rockford, USA) system fitted with a Q-Exactive quadrupole-Orbitrap mass spectrometer with a heated electrospray ionization (HES II) source. The electrospray ionization (ESI) conditions were as follows: sheath gas 40 arb; auxiliary gas 10 arb; ion spray voltage 3000–2800 V; temperature 350 °C; ion transport tube temperature 320 °C. The scanning mode was Full-ms-ddMS2 mode with positive/negative ion. Primary scan range (scan m/z range): 70–1050 Da, secondary scan 200-2000, primary resolution 70,000, secondary 17,500. The mass spectrometry instrument was externally calibrated with Pierce® ESI negative ion calibration solution (Thermo Scientific, Rockford, USA) and Pierce® LTQ VELOS ESI positive ion calibration solution (Thermo Scientific, Rockford, USA).

Reverse separation was performed using a C18 column (100 mm × 2.1 mm I.D., 1.8 μm, Waters HSS T3, Milford, USA). The chromatographic conditions were as follows: flow velocity of 0.3 mL min$^{-1}$, sample injection volume of 2 μL, column temperature of 40 °C, mobile phase A comprised of water containing 0.1% formic acid, and mobile phase B comprised of acetonitrile containing 0.1% formic acid. The elution gradient started at 1% v/v mobile phase B for 1 min and then linearly increased to 99% v/v B at 18 min. The mobile phase content was held at 90% v/v B for 4 min, decreased to 5% v/v B, and equilibrated at 5% v/v B for 4 min. The total analysis time was 20 min.

For chemical standard validation, quercetin, hyperoside, scopoletin and syringaldehyde analytical standards (purity>98%, J&K Scientific, CA, USA) were dissolved in methanol (1 mg mL$^{-1}$, mother solution), respectively. Then each solution was diluted to 10 μg mL$^{-1}$ using 50% acetonitrile. Each dilution was filtered through a polytetrafluoroethylene filter for UHPLC-MS/MS analysis as mentioned above. The platform of non-targeted metabolism and targeted validation work received technical support from Shanghai Sanshu Biotechnology Co., LTD.

## Isolation and growth of rhizosphere bacteria with rhizosphere metabolites

To determine whether the metabolites that were enriched in the peanut rhizosphere of PM-R (quercetin, hyperoside, scopoletin and syringaldehyde) selectively affected bacterial growth rates, we first isolated bacteria from PM-R peanut rhizosphere soil. Briefly, 1 *g* of fresh soil was $10^{-6}$ diluted with sterile water and vortexed at 3000 rpm for 2 min. Then, 100 μL of the soil suspension was spread on LB plates (tryptone 10 g L⁻¹, yeast extract 5 g L⁻¹, NaCl 10 g L⁻¹, agar 15 g L⁻¹, pH = 7.0) and incubated in the dark at 28 °C for 24–48 h. Single colonies were purified and sequenced by BGI Corp. (Shenzhen, China) for strain identification. To determine bacterial $N_2$-fixing capacity, each pure isolate was cultured on Ashby nitrogen-free agar (glucose 20 g L⁻¹, $KH_2PO_4$ 0.2 g L⁻¹, $MgSO_4$ 0.2 g L⁻¹, NaCl 0.2 g L⁻¹, $K_2SO_4$ 0.1 g L⁻¹, $CaCO_3$ 5.0 g L⁻¹ agar 15 g L⁻¹, pH = 7.0) at 28 °C for 5 days, respectively[64]. Isolated colonies that grew well on these plates were considered capable of $N_2$ fixation.

Next, we selected 27 isolates (including four representative isolates (N4, OP14, N68 and N69), four $N_2$-fixing rhizobial bacteria (N43, N45, N47 and N59) and 19 other strains) for growth rate comparison with different metabolite additions in microplate incubations (Fig. S8a). Isolates were activated in liquid LB at 28 °C for 24 h. Bacterial cells of each isolate were washed twice and adjusted to $OD_{600} = 0.5$ using 1/5 TSB medium (tryptone 3 g L⁻¹, soytone 0.6 g L⁻¹, glucose 0.5 g L⁻¹, NaCl 1 g L⁻¹, $K_2HPO_4$ 0.5 g L⁻¹, pH = 7.0). Then 1 mL of bacterial suspension was diluted 10 times for 96-well microtiter plate cultivation. For metabolite addition experiment, 5 μL of bacterial suspension was transferred to each well with 100 μL 1/5 TSB media containing 5 μg mL⁻¹ of one of the four metabolites (sterile solutions of quercetin, hyperoside, scopoletin, or syringaldehyde). As controls (C), we used the same amount of bacterial suspension and TSB media but without added metabolites, to assess baseline bacterial growth. In total 10 microtiter plates containing 810 filled wells ((4 metabolites+1 control) × 27 isolates x 6 replicates) were cultivated at 28 °C for 3 days. During the incubation, the $OD_{600}$ values of each well were measured as bacterial biomass using Microplate Reader (iMark, Bio-Rad Laboratories, Hercules, CA, USA) at 12 h time intervals. Since the basal growth curve of most strains showed logarithmic growth between the 12th and 48th hour of incubation (Fig. S8b), we used the biomass estimates from these two-time points to calculate the effect of each metabolite on bacterial growth, relative to the controls, as the variation in bacterial growth rate (*V*):

$$V = \frac{BG'_{(48)} - BG'_{(12)}}{BG_{(48)} - BG_{(12)}}$$

in which $BG'_{(48)}$ and $BG'_{(12)}$ represent bacterial biomass at the 48th and 12th hour of incubation with metabolite addition; and $BG_{(48)}$ and $BG_{(12)}$ represent bacterial biomass at these time points in the controls (TSB medium only).

To determine whether the rhizosphere metabolites also affected the $N_2$ fixation capacity of the three representative isolates (N4, N68 and N69), we used the same microtiter plate cultivation method as described above, but replaced the LB and 1/5 TSB media with Ashby nitrogen-free liquids. After incubating for 48 h, the quantity of bacterial $N_2$-fixation was determined using total nitrogen colorimetry in liquid. Each treatment group included six replicates.

## Effect of rhizosphere metabolites on Bradyrhizobial colonization and root nodulation

To investigate which of the four $N_2$-fixing rhizobial isolates (N43, N45, N47 and N59; all affiliated with *Rhizobium* and *Bradyrhizobium*) could successfully colonize peanut roots and induce host nodulation, these isolates were exogenously inoculated to plant seedlings growing in soil. First, bacteria were incubated in YT medium (tryptone 16 g L⁻¹,

yeast extract 10 g L⁻¹, NaCl 5 g L⁻¹, pH = 7.0) at 28 °C with shaking at 180 rpm until $OD_{600}$ reached 0.5. Activated cells were washed twice with sterile water and cell suspensions were adjusted to $OD_{600} = 0.5$ for use as microbial agents before inoculation. Meanwhile, surface sterilized peanut seeds were germinated and transplanted to pots with sterilized soil (diameter = 10 cm; height = 8 cm, one seedling per pot). After 5 days, each seedling was inoculated by injecting 5 mL of microbial agent into the soil around the root (six replicate pots for each isolate). All plants were placed in a growth chamber at a temperature of 28–30 °C, 40–55% relative humidity, and a 12/12 h light/dark photoperiod. Pots were sealed and watered every 2 days. After 30 days, plant roots were collected for nodule detection.

Among the four rhizobial isolates tested, *Bradyrhizobium* N47 was the only one that induced host nodulation in the pot experiment described above. Therefore, we selected N47 for an inoculation experiment to assess the effects of specific rhizosphere metabolites on peanut root nodulation by this strain. Surface sterilized seeds were germinated and grown in 1/5 MS medium (Murashige & Skoog basic medium, Duchefa Biochemie, Haarlem, Netherlands) in cell culture flasks for 15 days. Then, 2 mL of N47 microbial agent including 5 μg mL⁻¹ of one of the four metabolites (quercetin; hyperoside; scopoletin; syringaldehyde) was injected around the peanut roots. Flasks were placed in a growth chamber under the same culture conditions as for the pot experiment described above. Plants with only water (C, control) or N47 microbial agent (Br) injection were processed identically. After 24 h, plant roots were collected for RNA extraction and qRT-PCR of symbiosis signaling genes. Each treatment group included six replicates.

To investigate whether the rhizosphere metabolites influenced the nodulation signaling of *Bradyrhizobium* N47, we also measured the expression of bacterial nod genes in this strain after metabolite exposure. Briefly, 100 μL of N47 suspension ($OD_{600} = 0.2$) was transferred to 2 mL YT media in culture tubes and incubated at 28 °C with shaking at 180 rpm for 48 h. Then, each culture tube received one of the four metabolites (quercetin; hyperoside; scopoletin; syringaldehyde) to a final concentration of 0.5 μg mL⁻¹. The mixtures were cultured under the same conditions for another 48 h. The bacterial cells were collected for RNA extraction and qRT-PCR. Controls (C) of *Bradyrhizobium* N47 with water addition were processed identically. Each treatment group included six replicates.

## Nodule histological observation and construction of green fluorescent protein (GFP) –marked *Bradyrhizobium* N47 for root colonization

For histological observations of N47-induced nodules, the nodules were collected 30 days after N47 inoculation and fixed in a mixture of 2.5% glutaraldehyde and 4% paraformaldehyde in 0.1 M sodium cacodylate (pH 7.2). The fixed nodules were then dehydrated with ethanol and embedded in paraffin. Thin sections (5 μm) were stained with toluidine blue and viewed with a light microscope[65].

To observe the colonization of *Bradyrhizobium* N47 within the host root, we constructed a GFP-marked strain N47. The *Escherichia coli* DH 5α with GFP plasmid pJZ383 (Tet^S, GFP, Spe^R) were kindly provided by Prof. Zengtao Zhong (Life Science College, Nanjing Agricultural University, Nanjing, China). The wild strain N47 can grow on tetracycline (Tet^R, 5 μg mL⁻¹) TY agar media but is not resistant to spectinomycin (Spe^S, 100 μg mL⁻¹). The *E. coli* DH 5α carrying plasmid pJZ383 was conjugated with *Bradyrhizobium* N47 by biparental patch mating[66,67]. Transconjugants were selected on YT plates containing Tet and Spe. Transconjugants were purified by subculturing three times on the same medium and finally observed using a hand-held lamp (LUYOR-3415RG, Luyor Instrument, CA, USA) with excitation and emission wavelengths of 440 and 500 nm, respectively. Then, 2 mL GFP-marked N47 microbial agent was injected around the 20-day-old peanut roots in flasks. After 3 days of incubation in the growth

chamber, plant roots were collected and imaged using laser confocal scanning microscopy (CLSM710, Zeiss, Oberkochen, Germany) at 488 nm.

## Plant RNA extraction, cDNA library and transcriptome sequencing

Total RNA was extracted using the RNA isolation kit (E.Z.N.A. Total RNA kit for plant, Omega, Guangzhou, China; Spin Column Total RNA kit for bacteria,Sangon Biotech, Shanghai, China) following the manufacturer's protocol. RNA purity and quantification were evaluated using a NanoDrop 2000 spectrophotometer (Thermo Scientific, USA). RNA integrity was evaluated using an Agilent 2100 Bioanalyzer (Agilent Technologies, Santa Clara, CA, USA). Samples with RNA integrity number (RIN) > 7 were subjected to subsequent analysis. The libraries were constructed using the TruSeq Standed mRNA LTSample Prep Kit (Illumina, San Diego, CA, USA). Transcriptome sequencing and analysis were conducted by OE Biotech Co., Ltd. (Shanghai, China).

The libraries were sequenced on the Illumina sequencing HiSeqTM 2500 platform, and 150 bp paired-end reads were generated. Raw data (643.36 MB raw reads) in fastq format were processed using Trimmomatic[68], and the low-quality reads were removed to obtain clean reads (635.46 MB). Then, the clean reads were mapped to the peanut genome database (peanutbase:https://www.peanutbase.org/data/public/Arachis _hypogaea/Tifrunner.gnm2.J5K5/arahy.Tifrunner. gnm2.J5K5.genome_main.fna.gz) using hisat2[69]. The FPKM value of each gene was calculated using cufflinks[70], and the read counts of each gene were obtained by HTSeq-count[71]. Genes with differential expression were identified using the DESeq R package[72]. A q value < 0.05 and a fold change > 2 were set as the thresholds for significant differential expression. KEGG pathway enrichment analysis of DEGs was performed using R based on the hypergeometric distribution.

## Quantitative reverse transcription polymerase chain reaction (qRT-PCR)

Plant and microbial total RNAs were extracted using RNA isolation kit described above. RNAs were reverse transcribed into cDNA with a cDNA Synthesis Kit (Thermo Fisher Scientific, MA, USA) following the manufacturer's instructions. qRT-PCR was performed in a 10 μL PCR volume using a CFX Connect Real-Time System (Thermo Fisher Scientific, MA, USA) and TB GreenTM Premix Ex TaqTM (Takara, Kusatsu, Japan). All the qRT-PCR primers are listed in Supplementary Data 10, 11[73-88]. The peanut Actin gene was used as the internal control. The relative quantitation of target gene expression was calculated using the $2^{-\Delta\Delta Ct}$ method.

## Bacterial isolates purification and identification

Single colonies that appeared on the LB plate were picked and purified using the streaking method. Pure isolates were cultured in LB broth at 28 °C in a shaker rotating at 180 rpm for 48 h. Bacterial cells were washed and collected through centrifugation at 10, 000 rpm for 3 min for DNA extraction. Bacterial DNA was extracted using a DNA Kit (Omega Bio-Tek, Inc., Norcross, GA, USA) following the manufacturer's instructions. The universal primers 338 F and 806 R, consistent with those used for bacterial high-throughput sequencing, were used for PCR. Each sample was amplified in a 20 μL reaction system, which contained 0.5 μM forward and reverse primers, 1 × PremixTaq DNA polymerase (Takara, Kusatsu, Japan) and 20 ng DNA templates. After an initial denaturation at 95 °C for 3 min, the targeted region was amplified by 20 cycles of 94 °C for 30 s, 58 °C for 30 s, and 72 °C for 30 s, followed by a final extension at 72 °C for 1 min in a thermal cycler (GeneAmp PCR system 2700; Applied Biosystems, New York, USA). The PCR products were then purified using a DNA Gel Extraction Kit (Axygen Bioscience, Inc., Union City, CA, USA) and sequenced by BGI Corp. (Shenzhen, China) for strain identification.

## Stimulation of coumarin on the root nodulation of *Medicago truncatula*

The model microorganism *Sinorhizobium meliloti* strain1021 was activated in Yeast extract and Tryptone medium (YT, tryptone 16 g L⁻¹, yeast extract 10 g L⁻¹, NaCl 5 g L⁻¹, pH=7.0). Bacterial cells were washed twice using sterilized water and cell suspensions were adjusted to 0.03 at $OD_{600}$ for use as microbial agent[73].

To investigate whether the rhizosphere metabolites influenced the nodulation signaling of *S. meliloti* strain1021, we measured the expression of bacterial nod genes in this strain after metabolite exposure. Briefly, 100 μL of bacterial suspension (OD600 = 0.5) was transferred to 5 mL YT media in culture tubes and incubated at 28 °C with shaking at 200 rpm for 2 h. Then, each culture tube received one of the four metabolites (quercetin, hyperoside, scopoletin and syringaldehyde) to a final concentration of 0.1 μg mL⁻¹. The mixtures were cultured under the same conditions for another 12 h. The bacterial cells were collected for RNA extraction and qRT-PCR of Nod genes. Controls (C) of *S. meliloti* with water addition were processed identically. Each treatment group included six replicates.

Seeds of *M. truncatula* (A17) were sterilized using sulfuric acid and then germinated on 1% water agar medium at 4 °C for three days and 25 °C for one day[73]. Plant seedlings were transplanted to pots (length × width × depth = 10 × 10 × 10 cm) with vermiculite for growth (chamber with a 16 h light/8 h dark period at 22 °C; photon flux density = 250 μmol m⁻² s⁻¹). After 7 days, three treatments were set up to compare the effect of coumarin scopoletin on *M. truncatula* nodulation: (1) SM, seedling was injected with 5 mL microbial agent; (2) SML, seedling was injected with 5 mL microbial agent containing 5 μg mL⁻¹ scopoletin; (3) SMH, seedling was injected with 5 mL microbial agent containing 50 μg mL⁻¹ scopoletin. Each treatment was conducted 12 replicates. Controls with water injection were processed identically. Pots were placed back in the chamber. After 3 days of inoculation, 6 plant replicates of each treatment were collected for root RNA extraction and qRT-PCR determination. Other 6 replicates were collected for root nodule calculation and biomass detection after 30 days of inoculation. During plant growth, pots were sprayed with water every 5 days.

## Statistical analyzes

The data (over three treatment groups, follows a normal distribution) of field physiological traits, soil physicochemical properties, microbial alpha-diversity and qPCR of genes involved in rhizobial colonization, plant defense and nodulation in peanut and Medicago that were analyzed using Tukey's post-hoc tests for multiple comparisons to explore the significance of the difference between pairs of the treatments when One-way analysis of variance (ANOVA) terms were significant. For field experiment data, we calculated the mean of all samples (n = 3) for each plot to ensure a random selection for ANOVA analysis. Therefore, the replicates for each field treatment were three instead of nine. The data of (two treatment groups, follows a normal distribution) root transcriptome and gene expressions between PP and PM-R two treatment groups, enrichment of the four typical metabolites (quercetin, hyperoside, scopoletin, and syringaldehyde), isolates' growth rates were analyzed using two-tailed unpaired t-tests according to F-test results to compare significant differences. The μeast squares regression was performed to detect the relationships between root nodulation, soil ¹⁵N₂ fixation, plant biomass and rhizosphere nitrogen (including total, ammonia and nitrate nitrogen). The assumptions of homoscedasticity and normality of the residuals of the regression models were checked to confirm the reliability of the models. The hierarchical clustering based on Euclidean distance and heatmaps of metabolites and transcriptional functional genes were performed with the "heatmap" package[89]. Principal Component Analysis (PCA) and similarity analysis (ANOSIM) of metabolome and transcriptome data based on the pairwise Euclidean distance were performed to reveal the dissimilarities among different cropping treatment groups using the "stats"

package[90]. Multivariate homogeneity of groups dispersions of bacterial amplicon sequencing data based on Bray–Curtis distance was conducted using "vegan" package[91]. Ternary plots for identifying specific enriched microbial taxa in the cropping treatment groups were generated with the "vcd" package[92].

## Reporting summary

Further information on research design is available in the Nature Portfolio Reporting Summary linked to this article.

## Data availability

The 16 S rRNA gene sequences and peanut transcriptome sequences were deposited into the Sequence Read Archive (RDA) of the National Centre for Biotechnology Information (NCBI) database under the accession numbers PRJNA718421 and PRJNA790729. The sequences of all isolates (including deleted duplicate data) were deposited into the National Centre for Biotechnology Information (NCBI) database under accession number MW856489-MW856613. Raw nontarget metabolite data were deposited into the European Molecular Biology Laboratory (EMBL-EBI) MetaboLights database under accession number MTBLS6537 (www.ebi.ac.uk/metabolights/ MTBLS6537). Source data are provided with this paper.

## Code availability

The authors declare that the R (R 4.3.1) codes used to generate the results reported in this study are available in this paper. The R code supporting the findings presented here is available from the GitHub Repository (https://github.com/Pop-rainbow/paper/blob/main/NCOMMS%2023-15056%20Code%20file.docx).

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

## Acknowledgements

We thank J. Fan, X Liu, L. Chen, Y. Jiang, L. Zhu and X. Wang (Institute of Soil Science, Chinese Academy of Sciences) for their assistance in the historical management of field experiments. We thank Y. Liu (Institutional Center for Shared Technologies and Facilities of NIGLAS, CAS) for the assistance in $^{15}$N isotope measurement, H. Chen (Soil & Environment Analysis Center, CAS) and Shanghai SanShu Biotechnology Co., LTD for the assistance in UHPLC-MS/MS, J Yang (Center for Excellence in Molecular Plant Sciences, Chinese Academy of Sciences) and L Luo (School of Life Science Shanghai University) for the Madicago growth and *S. meliloti* molecular analysis. We thank C. Ettema from the Netherlands for her through editing of the manuscript. We also thank Prof. Nico Eisenhauer (German Centre for Integrative Biodiversity Research, iDiv, Germany), Prof. Jos Raaijmakers (Netherlands Institute of Ecology, NIOO-KNAW, Netherlands) and Prof. Anna H. Buschart (University of Amsterdam, Netherlands) for their suggestions on the review comments and data statistics. We sincerely appreciate the support and encouragement provided by our international and domestic colleagues in the scientific research community to the corresponding author Yan, following the unexpected passing of professor Bo Sun, the team leader. This study was financially supported by the Science Foundation of the Chinese Academy of Sciences [grant number ISSASIP2211], the National Key R&D Programs [grant number 2022YFD1900601 and 2023YFD1900201], the National Natural Science Foundation of China [grant number 41977098].

## Author contributions

Y.C., M.B. and B.S. collaboratively designed the experiment and discussed conceptual ideas. Y.C., M.Q., R.S., Z.W., C.D., X.G., K.D. and E.W. actively participated in a series of field and laboratory experiments. R.S., Y.C., M.Q., K.D., C.D. and J.Z. contributed to data curation and analysis. Y.C., M.Q., R.S., Z.W., M.B. and B.S. were involved in the creation of the original draft. Y.C., M.Q., R.S., M.B., J.Z., X.P. and E.W. participated in reviewing and editing each round of manuscript revision.

## Competing interests

The authors declare no competing interests
