## [Peer Review File · Nature Communications]

Reviewers' Comments:

Reviewer #1:

Remarks to the Author:

In this manuscript, Chen et al. aim to understand the influence of neighboring plants on rhizosphere metabolite composition and the resulting changes in the microbiome for soil biological nitrogen fixation. They utilize a unique long-term experiment where peanuts were grown for 8 years under monoculture (designated PP), rotation with Oilseed rape (P-R) and intercropping with maize rotated with rape (PM-R). Field samples were collected for the analysis of rhizosphere metabolome, root transcriptome and microbiome compositions. As a follow-up to this analysis, the authors isolated a number of rhizosphere microbes that were associated with the PM-R treatment and used them to explore how and why flavonoids and coumarin influence the biological nitrogen fixation including free-living and symbiotic N₂ fixation. This is a truly unique dataset in terms of omics. Further, the topic is clearly hot and fascinating: how does intercropping drive the increase in plant productivity and resource use efficiency (nitrogen in particular) that has been reported regularly to occur in these mixed cropping systems. The authors give a clear chemical clue to assess the underlying mechanism for the increase of N availability with crop diversification

I have some suggestions for the authors to consider, some of which are major.

The authors stated that biological nitrogen fixation can be improved by free-living bacteria and rhizobial symbiosis with legumes. However, data at the beginning of the result (Fig.1) just highlight the clue of symbiotic N₂ fixation (peanut nodulation to soil ammonia/nitrate/total nitrogen). There is no direct clue to support the importance of free-living bacterial N₂ fixation in soil N availability. To make better logical coherence, detailed data for free-living nitrogen fixation should be added, such as the improvement of soil nitrogen components.

The effect of neighboring maize on N-fixation by bacteria is interesting. But what is the role of oilseed rape, as it is another important crop in the treatments? It seems that oilseed-rape rotation also increased the nodule density, but resulted in low biomass. Is that mean the negative plant-soil feedback of peanut-oilseed rape rotation? Although the legacy effect of oilseed-rape planting was added in Fig.6, there isn't any discussion in the main text. The authors should give some explanation to make it clear: why is oilseed rape selected for the crop diversification experiment?

The long-term field plots were sampled at a single time point, then how to eliminate the differences in soil nitrogen concentration caused by the historical accumulation of different levels of exogenous nutrients? For example, extra fertilizer is added in winter for oilseed rape planting every year. This may influence the status of soil nitrogen.

Fig 2 and 3 can be merged after moving unimportant and descriptive figures (such as Fig.2B and Fig. 3E) to supplementary information. This allows the topic of this paragraph to focus more on the discovery of important metabolites.

It is too normal to show the bacterial diversity and composition survey in Fig.4. Actually, the authors have found the PM-R enriched OTUs. Whether the bacterial enrichment is a consequence of the accumulation of specific metabolites (quercetin, hyperoside, Scopoletin and Syringldehyde)? It can be confirmed by linear correlation analysis.

The abbreviations in Fig.4C (PPr, P-Rr and PM-Rr) do not match the legend (PP, P-R and PM-R).

Reviewer #2:

Remarks to the Author:

In this article, the authors use various techniques, including metabolic analysis and high-throughput sequencing, to investigate the impacts of rotation and intercropping of legume peanut.

The authors of this article go a little more in-depth on the interaction between plants and their rhizobiome and, despite this topic is not really innovative and it doesn't really get really impressive

results. I believe that understanding this connection is essential for develop agricultural advancements.

However, before considering publishing this manuscript, I think there are a few issues that need to be addressed.

major comments:

1°.lines 388-390. The description of the experimental design is not clear for me. I understood this design includes pseudoreplicates, therefore, these samples could not be used in an analysis of variance.

2°. In my opinion, the processing of sequencing samples is outdated. According to the MS, the tool employed was QIIME (line 431), which was which was replaced by qiime2 in 2018 and it doesn't have support. According to the MS the samples were collected on June 2019, therefore the authors could update the bioinformatic process.

3°. why OTU instead of ASV?. According to the comparisons, a greater diversity can be obtained using ASV. Perhaps, using ASV you could have obtained a higher phylogenetic consistency (Line 210-211)

4°. Similar to the point 2, why did you use silva v132?. Silva 138 was released on December 2019.

5°. The statistics in this post are its weakest element, in my opinion. The authors state that a one-way anova was performed to compare management differences, although it is never stated whether the assumptions for this parametric technique were satisfied. This makes me doubt the results obtained. The same for the t.test.

6°. Why did you choose a non-parametric method to study the differences between managements instead of the PERMANOVA?

7° Please, cite the r packages according to the authors.

8°. In the figure 1 the authors explain that they performed an OLS regression. why was this not include in material and methods?. Were the assumptions tested?

minor comments:

9° Some parts should be removed or changed from results, for example, I consider than lines 113 to 117 should be in material and methods.

10° Line 128. is log fold change or log 2 fold change?. Please check, because the table indicates a log fold change.

11° ANOVA followed by t-test?, why?. please clarify.

12. I think the figures are not displayed correctly.

I hope these comments can help you to improve the manuscript, I wish you luck.

Reviewer #3:

Remarks to the Author:

The authors demonstrated that coexisting crops influence nitrogen fixation in legumes. They have used a field study (8-year-old field experiment) and provided clear evidence that peanut plants co-

cultured with maize (and with oilseed rape rotation) induce specific changes in the microbiome and metabolite composition of the rhizosphere of peanuts, as well as modulation of expression specific genes in peanut roots. Further, the authors proved that enriched metabolites (flavonoids and coumarins) affect positively nitrogen fixers' root colonization and activity. The findings of this work showcase profound insights to increase nitrogen availability in sustainable agriculture. While the authors delivered rigorous experimental design and proved that flavonoids and coumarins were detected only in the rhizosphere of peanut co-cultured with maize, they have not unraveled the mechanism of interspecific interactions between peanut and maize that leads to the enrichment of these metabolites. The latter could be a plant-plant interaction (possibly maize root exudates inducing changes in the peanut rhizosphere) or maize microbiome could also induce changes in the peanut rhizosphere. The mechanism of this co-existence is still not elucidated and would be an outstanding finding as compared to the solid understanding of the effect of flavonoids on nitrogen-fixing bacteria. The figures are of terrible quality.

Major comments:

1-I highly recommend confirming the identification of the metabolites tested (flavonoids and coumarin) using appropriate measurement of a reference standard with MS, MS/MS, and retention time matching.

2-Table S2: change confidence level to mzCloud score. Please add the following information, it is a list of random metabolite names otherwise:

a.Neutral mass (Da)

b.Mass Detected ES(-) and/or ES(+)

c.Relative mass defect (ppm)

d.RT [min]

e.Fragment ions

3-Lines 118-119. Why have you set the mzCloud Score Match to 70? The list of compound identification should pass through a manual curation to confirm the identification and confidence level. Please check the standard protocol for compound identification – Schymanski et al, Environmental Science & Technology 2014 48 (4), 2097-2098.

Minor comments:

1.line 74: I would change "aboveground diversity" to "plant diversity", as roots of both maize and peanut are part of the belowground.

2.Figs: 2B, 3D, 5, S2 are of poor quality, suggest to make them much bigger.

3.Fig. S2B. Spectra are small and I cannot see the numbers for checking the molecular ions and fragments. Chemical structure is almost transparent.

4.Lines 293-295: Provide a reference.

5.Lines 352-354. Could you try Mendicago transgenic plants to test the effect of coumarins influence nodulation?

6.Methods, line 443: extracted in methanol or in acetonitrile (as describe in the supplemental material)?

Point-by-point responses to comments from Reviewers

(Legume rhizodeposition of flavonoids and coumarin promotes nitrogen fixation by soil microbiota under crop diversification, Manuscript ID: NCOMMS-23-15056)

Reviewer 1 (Remarks to the Author):

In this manuscript, Chen et al. aim to understand the influence of neighboring plants on rhizosphere metabolite composition and the resulting changes in the microbiome for soil biological nitrogen fixation. They utilize a unique long-term experiment where peanuts were grown for 8 years under monoculture (designated PP), rotation with Oilseed rape (P-R) and intercropping with maize rotated with rape (PM-R). Field samples were collected for the analysis of rhizosphere metabolome, root transcriptome and microbiome compositions. As a follow-up to this analysis, the authors isolated a number of rhizosphere microbes that were associated with the PM-R treatment and used them to explore how and why flavonoids and coumarin influence the biological nitrogen fixation including free-living and symbiotic N₂ fixation. This is a truly unique dataset in terms of omics. Further, the topic is clearly hot and fascinating: how does intercropping drive the increase in plant productivity and resource use efficiency (nitrogen in particular) that has been reported regularly to occur in these mixed cropping systems. The authors give a clear chemical clue to assess the underlying mechanism for the increase of N availability with crop diversification. I have some suggestions for the authors to consider, some of which are major.

Re: Thank you for your strong encouragement and insightful comments. Based on your following opinions, we carefully considered them and made corresponding changes. Please see our point-by-point responses below.

The authors stated that biological nitrogen fixation can be improved by free-living bacteria and rhizobial symbiosis with legumes. However, data at the beginning of the result (Fig.1) just highlight the clue of symbiotic N₂ fixation (peanut nodulation to soil ammonia/nitrate/total nitrogen). There is no direct clue to support the importance of free-living bacterial N₂ fixation in soil N availability. To make better logical coherence, detailed data for free-living nitrogen fixation should be added, such as the improvement of soil nitrogen components.

Re: Thank you for your suggestion. In the original version, we placed the soil properties (such as rhizosphere ammonia and nitrate nitrogen concentration) in the supplementary information (Table S1). This may lead the readers to spend time flipping through the data from files to connect the logic of the writing. With the inspiration of the reviewer's comment, we added a new experiment: cultivate soil using ¹⁵N gas nitrogen for one week. Then, we measured the concentration of ¹⁵N fixed in soil. This result is a direct evidence to support the importance of free-living bacterial N₂ fixation in soil N availability. The result was shown in revised Fig. 1. Obviously, the highest crop diversity (PM-R) resulted in the highest soil ¹⁵N fixation. Therefore, we added the following sentences: "With respect to the rhizosphere free living N₂ fixation, microbial immobilization of molecular ¹⁵N was measured using inoculation of ¹⁵N₂ isotope labelling. Obviously, soil δ¹⁵N in PM-R was the highest, with 16% and 4% higher (p<0.05) than PP and P-R after 7 days of incubation, respectively (Fig. 1B). Plant biomass (p=0.127), rhizosphere ammonia and nitrate nitrogen levels were positively correlated with soil ¹⁵N fixation, but only nitrogen components were significant (Fig. 1D and F; p<0.001)." (Line 90-96, page 3-4)

Fig. 1 Effect of crop diversification on root nodulation, peanut growth, and rhizosphere ^{15}N fixation. (A) Effect on peanut nodulation. (B) Effect on peanut rhizosphere ^{15}N fixation. The data in A-B are shown as the mean \pm standard deviation ($n=9$). The error bars with lowercase indicate significant differences between groups ($p<0.05$) via one-way ANOVA and Tukey's post-hoc tests. (C) Correlations between root nodulation and peanut plant biomass. (D) Correlations between soil ^{15}N fixation and plant biomass. (E) Correlations between root nodulation and rhizosphere nitrogen components including total (black points with regression), ammonium (blue points with regression) and nitrate (green points with regression) nitrogen. (F) Correlations between soil ^{15}N fixation and rhizosphere nitrogen components including total, ammonium and nitrate nitrogen. Lines represent the least squares regression fits and shaded areas represent the 95% confidence intervals. PP, P-R and PM-R represent peanut monocropping, peanut-oilseed rape rotation, and peanut-maize intercropping rotated with oilseed rape, respectively.

The effect of neighboring maize on N-fixation by bacteria is interesting. But what is the role of oilseed rape, as it is another important crop in the treatments? It seems that oilseed-rape rotation also increased the nodule density, but resulted in low biomass. Is that mean the negative plant-soil feedback of peanut-oilseed rape rotation? Although the legacy effect of oilseed-rape planting was

added in Fig.6, there isn't any discussion in the main text. The authors should give some explanation to make it clear: why is oilseed rape selected for the crop diversification experiment?

Re: This is a very good question, Thanks! Yes, compared with the monocropping, peanut/oilseed rape rotation increased the leguminous nodules, but have a negative influence on peanut biomass (Fig. S2). The reason for the on-site investigation was a number of peanuts developed disease (increased black disease spots on plant leaves), which inhibited plant growth. This negative plant-soil feedback of peanut-oilseed rape rotation has been brought to our attention and a related study on the mechanisms is being prepared. However, diversification in regulating plant immunity is another important topic, beyond the scope of this current report.

Based on your suggestion, we described the results “Although peanut biomass was lower in the rotation system (P-R) than in the monoculture system ($p < 0.05$), this did not affect fruit yield ($p > 0.05$).” (Line 80-82, page 3). Meanwhile, we added a short discussion “Unlike the neighboring maize effect, peanuts that were rotated with oilseed rape did not result in biomass promotion despite increasing nitrogen fixation in soil, suggesting that, besides resource availability, there may be other factors in the legacy effect derived from historical species determining peanut development^{18,19,22}.” (Line 302-305, page 10)

Oilseed rape is a crop commonly grown in winter in tropics and subtropics. To make our basic theoretical research to support local practices, we selected oilseed rape as one of diversified crops in this study. We have added the explanation in M&M: “The field experiment compared three crop systems based on common practices in local intensive agricultural systems of tropics and subtropics^{38,51}.” (Line 386-387, page 12)

The long-term field plots were sampled at a single time point, then how to eliminate the differences in soil nitrogen concentration caused by the historical accumulation of different levels of exogenous nutrients? For example, extra fertilizer is added in winter for oilseed rape planting every year. This may influence the status of soil nitrogen.

Re: Thank you for your question! When we collected peanut rhizosphere soil of different treatments, we also collected corresponding bulk soil samples. By comparing nutrient differences in rhizosphere and bulk soils, we could clarify the nutrient contribution of historical nutrients in each treatment (Table S1). Although a small amount of fertilizer is applied during winter rape planting, this did not cause an increase in the bulk of P-R and PM-R treatments. This may be because of the rapid turnover of fertilizer nutrients due to climatic factors (high temperature and rain) in the subtropical area.

Fig 2 and 3 can be merged after moving unimportant and descriptive figures (such as Fig.2B and Fig. 3E) to supplementary information. This allows the topic of this paragraph to focus more on the discovery of important metabolites.

Re: Thanks for your nice suggestion. We've merged Fig.2 and Fig.3 in the revised manuscript. (Fig. 2)

It is too normal to show the bacterial diversity and composition survey in Fig.4. Actually, the authors have found the PM-R enriched OTUs. Whether the bacterial enrichment is a consequence of the accumulation of specific metabolites (quercetin, hyperoside, Scopoletin and Syringldehyde)? It can be confirmed by linear correlation analysis.

Re: Based on reviewer's comment, we revised Fig.4 and did the linear correlation analysis between each PM-R enriched ASVs and specific metabolites (Based on another reviewer's requirement, OTUs from the high-throughput sequencing were converted to ASVs data). The majority of enriched ASVs are positively correlated with one or more metabolites ($p < 0.05$), therefore we used heatmap (Fig. 3E) to show these correlations clearly as follow:

Fig. 3 Effect of crop diversification on the peanut rhizosphere bacterial community. (A) Shannon and Chao1 richness indices. The error bar with lowercase indicates significant difference between groups ($p < 0.05$) via one-way ANOVA and Tukey's post-hoc tests. (B) Principal coordinate analysis (PCoA) of bacterial beta dispersion among different samples based on Bray–Curtis distance (left) and distance of centroid beta-dispersal values for groups (right). Black lines indicate the median values. P values were adjusted using multiple (95% family-wise confidence level) comparisons using Tukey's HSD. (C) Phylum-level distribution of ASVs. (D) Ternary plot of bacterial ASVs shared among the different peanut rhizosphere communities. Circle sizes represent the relative abundances of the bacterial ASVs identified. The grey circles represent ASVs of which the relative abundance did not differ significantly between crop systems; the dark red, orange and green circles represent ASVs that had significantly higher relative abundances in the peanut rhizospheres in peanut monocropping (PP), peanut-oilseed rape rotation (P-R), and peanut-maize intercropping rotated with oilseed-rape (PM-R), respectively. The green circles with red border represent ASVs with the same marker sequence as the subsequent bacterial isolates (Fig.4A). (E) Heatmap of specific enriched metabolites and PM-R enriched bacterial ASVs according to Pearson's correlations. Positive and negative correlations are shown in blue and red, respectively. * $p < 0.05$, ** $p < 0.01$, *** $p < 0.001$.

The abbreviations in Fig.4C (PPr, P-Rr and PM-Rr) do not match the legend (PP, P-R and PM-R).

Re: We do apologize for our mistakes in labeling. We corrected them in the revised MS (Fig. 3B).

Reviewer 2 (Remarks to the Author):

In this article, the authors use various techniques, including metabolic analysis and high-throughput sequencing, to investigate the impacts of rotation and intercropping of legume peanut.

The authors of this article go a little more in-depth on the interaction between plants and their rhizobiome and, despite this topic is not really innovative and it doesn't really get really impressive results. I believe that understanding this connection is essential for develop agricultural advancements.

Re: First, thank you for the reviewer's precious time in our manuscript. Previous ecological studies have shown that leguminous plant diversification could increase the accumulation of soil nitrogen. To answer "Why and where does this increased nitrogen come from?", more studies simply attribute it to historical legume planting (the return of nitrogen-rich residues of legumes), but less known about how legume self-responds to plant diversification. Although Li et al (2016) found that flavonoids in the intercropping system promoted legume nodulation, the exact origin of the flavonoids is not clear (The authors use exogenous addition of genistein to simulate legume flavonoid production. However, genistein itself is a kind of flavonoid. The causal relationship here is not clear because of the limitation of research technology at that time). Our manuscript confirmed that the two chemical cues (not only flavonoids but also coumarin) were produced by the legume itself when confronted with historical and neighboring plant species using multi-omics and molecular biology methods. These metabolites could modulate two different types of nitrogen fixation patterns (1) for free-living N₂ fixers: metabolites selectively activate the growth of microorganisms to improve N₂ fixation; (2) for symbiotic N₂ fixers: these metabolites act as signals to enhance microbial colonization and root nodulation for N₂ fixation.

In addition, another important finding is that coumarin (secreted in the diverse cropping system) has the potential to boost plant defenses when promoting legume nodulation. It at least has been verified by the model plant *Medicago*. It seems to break the traditional concept of individual plant trade-off between growth and defense. This win-win model for legume growth and defense promotion is based on the plant-microbial alliance built by chemical signaling.

As for how these metabolites (represented by flavonoids) initiate legume symbiotic nitrogen fixation, it has been well explored by plant molecular biologists (Oldroyd et al., 2013, 2022; Yang et al., 2022). Here, we attempt to combine reductionism-based molecular botany with community (above- and below-ground) ecology to reveal the common ecological phenomena of plant-microbe interaction. It is a challenging beginning and we thank the reviewers for giving us the opportunity to address such challenges.

References:

- [1] B. Li, Y. Y. Li, H. M. Wu, F. F. Zhang, C. J. Li, X. X. Li, H. Lambers, L. Li, Root exudates drive interspecific facilitation by enhancing nodulation and N₂ fixation. *Proc. Natl. Acad. Sci. U.S.A.* 113, 6496-6501(2016).
- [2] G. E. D. Oldroyd. Speak, friend, and enter: signaling systems that promote beneficial symbiotic associations in plants. *Nat. Rev. Microbiol.* 11: 252-263 (2013).
- [3] G.E.D. Oldroyd, O. Leyser. A plant's diet, surviving in a variable nutrient environment. *Science* 368: eaba0196 (2020).
- [4] J. Yang, L.Y. Lan, Y. Jin, N. Yu, D. Wang, E.T. Wang. Mechanisms underlying legume-rhizobium symbioses. *J. Integr Plant Biol.* 64:244-267 (2022).

However, before considering publishing this manuscript, I think there are a few issues that need to be addressed.

major comments:

1°.lines 388-390. The description of the experimental design is not clear for me. I understood this design includes pseudoreplicates, therefore, these samples could not be used in an analysis of variance.

Re: We are sorry that our description caused you confusion and misunderstanding. To make it clearer, I provide the following figure for the reviewer. In the field experiment, we have three treatments (PP, P-R and PM-R). Each treatment is conducted with three plots (3 replicates). The size of each plot is 100 m² (20m×5m) (upper left of the following Figure). We planned to randomly collect 3 soil replicates in each plot. Therefore, we would have 9 replicates for each treatment. Considering the large area of each plot but the small area of soil sampling (bulk soil 15cm×15cm, rhizosphere soil from three plants) would affect the uniformity of these representative samples, we improved our methods for sample collection (Bottom right of the following Figure): we divided the 100² area artificially into 3 subplots and collected 6 soil samples in an S-shape (shown by dotted lines) in each subplot. Then these 6 samples were mixed into a composite sample. Therefore, three composite samples were collected from each plot, and nine were collected from each treatment. Compared with randomly collected samples, our collections cover the entire area of plots and minimize soil heterogeneity due to sampling distance.

Figure The detail of field experiment design and sampling

2°. In my opinion, the processing of sequencing samples is outdated. According to the MS, the tool employed was *QIIME* (line 431), which was which was replaced by *qiime2* in 2018 and it doesn't have support. According to the MS the samples were collected on June 2019, therefore the authors could update the bioinformatic process.

Re: The methodological comments (comments 2-4) are very professional. Based on your second to fourth comments listed below, we re-filtered and re-analyzed the raw data from high-throughput sequencing. We used *QIIME 2* to replace *QIIME*. The details of the analysis method are modified as follows:

“The adapt and primer sequences were removed using *Cutadapt* (v4.4)⁵⁷. Then *DADA2* (Divisive

Amplicon Denoising Algorithm 2) (version 1.26) was used to merge, filter, trim, and denoise the raw data, and finally generate amplicon sequence variants (ASVs)⁵⁸. Taxonomic assignment for the ASVs was performed using RDP Classifier against the SILVA rRNA database (version 138)⁵⁹. After removing mitochondria and chloroplast, samples were rarefied to the same sequencing depth (16000) for further analysis.” (Line 449-454, page 14)

References:

57. M. Martin. Cutadapt removes adapter sequences from high-throughput sequencing reads. *Embnet J.* 17, 10-12 (2011).

58. B.J. Callahan, P.J. McMurdie, M.J. Rosen, A.W. Han, A.J. Johnson, S.P. Holmes, DADA2: High-resolution sample inference from Illumina amplicon data. *Nat. Methods* 13, 581-583 (2016).

59. Q. Wang, G. M. Garrity, J. M. Tiedje, J. R. Cole, Naive Bayesian classifier for rapid assignment of rRNA sequences into the new bacterial taxonomy. *Appl. Environ. Microbiol.* 73, 5261-5267(2007).

3°. *why OTU instead of ASV?. According to the comparisons, a greater diversity can be obtained using ASV. Perhaps, using ASV you could have obtained a higher phylogenetic consistency (Line 210-211)*

Re: Agree and thanks for your suggestion. ASV is considered more accurate than OTU in high-throughput sequencing data analysis. In the revised manuscript, the data from high-throughput sequencing were re-analyzed using DADA2 to generated ASVs, and all relevant analysis were re-performed based on the new dataset. The details of the modified methods were replied above (Comment No.2). Due to the new analysis method, there were obvious differences in the alpha-diversity of rhizosphere microbial community; while at the beta-diversity level, the community was almost consistent with the results of the previous analysis method (Fig. 3).

4°. *Similar to the point 2, why did you use silva v132?. Silva 138 was released on December 2019.*

Re: Thank you! Similar to the point 2, The details of the modified methods were replied above (Comment No.2). We used Silva138 for taxonomic assignment.

5°. The statistics in this post are its weakest element, in my opinion. The authors state that a one-way anova was performed to compare management differences, although it is never stated whether the assumptions for this parametric technique were satisfied. This makes me doubt the results obtained. The same for the t. test.

Re: Thanks for the reviewer’s comments on the statistical methods of our MS. When we sorted out the data, we did a normal distribution test in advance and then selected subsequent statistics based on the normal distribution results. But in the material and methods section, we ignored the description of these details. We are sorry for our carelessness making the reviewer confusing.

Here, we used Q-Q (quantile-quantile) plots as the example to assess whether or not a variable is normally distributed. The points lie mostly along the straight diagonal line with some minor deviations along each of the tails. Based on these plots, we assume that these set of data are normally distributed.

Figure Quatile-Quatile (Q-Q) plots for the normally distribution test of plant performance, soil nitrogen, soil bacterial diversity (A) and the concentration of key metabolites in peanut rhizosphere (B).

Meanwhile, in the revised manuscript, we added the detail as follow in M&M:

“The data (over three treatment groups, follows a normal distribution) of field physiological traits, soil physicochemical properties, microbial alpha-diversity and qPCR of genes involved in rhizobial colonization, plant defense and nodulation in peanut and Medicago that were analysed using Tukey’s post-hoc tests for multiple comparisons to explore the significance of the difference between pairs of the treatments when One-way analysis of variance (ANOVA) terms were significant. The data of (two treatment groups, follows a normal distribution) root transcriptome and gene expressions between PP and PM-R two treatment groups, enrichment of the four typical metabolites (quercetin, hyperoside, scopoletin, and syringaldehyde), isolates’ growth rates were analysed using two-tailed unpaired t-tests according to F-test results to compare significant differences. The least squares regression was performed to detect the relationships between root nodulation, soil $^{15}\text{N}_2$ fixation, plant biomass and rhizosphere nitrogen (including total, ammonia and nitrate nitrogen). The assumptions of homoscedasticity and normality of the residuals of the regression models were checked to confirm the reliability of the models.” (Line 577-591, page 18)

6°. Why did you choose a non-parametric method to study the differences between managements instead of the PERMANOVA?

Re: Sorry for the confusion. As we mentioned above, our dataset in the study were normally distributed, thus the parametric method was used to study the differences between managements. It has been corrected in the revised manuscript. The improved detail has been answered and listed above (comment 5)

7° Please, cite the *r* packages according to the authors.

Re: OK, done.

References

- 66. R. Kolde. Package“Pretty heatmaps”.Website: <https://rdocumentation.org/packages/heatmap/versions/1.0.12>).
- 67. J. Oksanen, F.G. Blanchet, M. Friendly, R. Kindt, P. Legendre, D. McGlinn, P. P. Minchin, R.B. O’Hara. Vegan: Community Ecology Package. R package version 2.5-4 (2019).
- 68. A. Field, J. Miles, Z. Field. Discovering Statistics Using R. London (Sage publications, 2012).
- 69. D. Meyer, A. Zeileis, K. Hornik, F. Gerber, M. Friendly. Vcd: visualizing categorical data. <https://CRAN.R-project.org/package=vcd> (2015). (Line 815-822, page 23)

8°. In the figure 1 the authors explain that they performed an OLS regression. why was this not include in material and methods?. Were the assumptions tested?

Re: Thank you for reviewer’s criticism and suggestions on the OLS regression analysis. When we read the relevant papers published in Nat Comms, we found that quite part of authors prefer to add the details of statistical methods in the figure legends. This may effectively reduce the word number in the main text. With your suggestions, the detail of least squares regression has been added in Material and Methods as follow:

“The least squares regression was performed to detect the relationships between root nodulation, soil ¹⁵N₂ fixation, plant biomass and rhizosphere nitrogen (including total, ammonia and nitrate nitrogen). The assumptions of homoscedasticity and normality of the residuals of the regression models were checked to confirm the reliability of the models.” (Line 587-591, page 18)

We listed the evidence of the homoscedasticity and normality of the residuals of the regression model below:

Figure Homoscedasticity and normality (Q-Q plots) of the residuals of the regression models in Fig.1C, D and F.

Figure Homoscedasticity and normality (Q-Q plots) of the residuals of the regression models in Fig.4C.

Figure Homoscedasticity and normality (Q-Q plots) of the residuals of the regression models in Fig.1E.

minor comments:

9° Some parts should be removed or changed from results, for example, I consider that lines 113 to 117 should be in material and methods.

Re: Thank you. According to your suggestion, the sentence “To determine the effect of crop diversification on metabolic deposition in the peanut rhizosphere...” has been moved to Material and Methods (Line 456-457, page 14). In addition, we shortened the beginning of the paragraph of “Crop diversification alters the peanut rhizosphere bacterial community” and “Crop diversity-enhanced rhizosphere metabolites trigger bacterial nitrogen fixation” as follow:

“Given the effects of crop diversification on peanut root metabolic biosynthesis and release (Fig. 2), we next investigated whether these changes in rhizosphere chemistry has the ripple-on effect on the root associated bacterial community.” (Line 164-166, page 6)

“As we expected, the rhizosphere bacterial community was selectively influenced by the production and exudation of active specific metabolites from peanuts in the system with maize and rape. However, it is not clear why peanut alters its rhizosphere chemistry and microbiota. It could be related to the higher plant nitrogen fixation exhibited in the field of PM-R group.” (Line 187-191, page 6)

10° Line 128. is log fold change or log₂ fold change?. Please check, because the table indicates a log fold change.

Re: We are sorry for our mistake. In fact, it is log₂ fold change. We’ve corrected it as “log₂ (Fold change)” in Fig. 2B.

11° ANOVA followed by t-test?, why?. please clarify.

Re: We are sorry for the vague and wrong description of the statistical method used in the original version. Based on your above comments (comment 5-11). We consulted with colleagues in eco-statistics and re-analysed the data. The improved detail has been answered and listed above (comment 5)

12. *I think the figures are not displayed correctly.*

Re: Thank you! Based on the other two reviewers' comments on the figures, some figures have been modified to meet all reviewer's requirements.

I hope these comments can help you to improve the manuscript,

I wish you luck.

Re: Thank the reviewer again for your time and for giving us the opportunity to revise the manuscript. We will try our best to improve the MS based on your nice comments. To be honest, this is also the best chance for us to deepen our understanding of this research through communication with top international professional scientists. You and the other two reviewers' comments actually give us a lot of inspiration to plan our future plant-microbe interaction in crop diversity ecosystems. Thank you!

Reviewer 3 (Remarks to the Author):

The authors demonstrated that coexisting crops influence nitrogen fixation in legumes. They have used a field study (8-year-old field experiment) and provided clear evidence that peanut plants co-cultured with maize (and with oilseed rape rotation) induce specific changes in the microbiome and metabolite composition of the rhizosphere of peanuts, as well as modulation of expression specific genes in peanut roots. Further, the authors proved that enriched metabolites (flavonoids and coumarins) affect positively nitrogen fixers' root colonization and activity. The findings of this work showcase profound insights to increase nitrogen availability in sustainable agriculture. While the authors delivered rigorous experimental design and proved that flavonoids and coumarins were detected only in the rhizosphere of peanut co-cultured with maize, they have not unraveled the mechanism of interspecific interactions between peanut and maize that leads to the enrichment of these metabolites. The latter could be a plant-plant interaction (possibly maize root exudates inducing changes in the peanut rhizosphere) or maize microbiome could also induce changes in the peanut rhizosphere. The mechanism of this co-existence is still not elucidated and would be an outstanding finding as compared to the solid understanding of the effect of flavonoids on nitrogen-fixing bacteria. The figures are of terrible quality.

Re: First of all, thanks for the reviewer's comments. As you mentioned above, there are still points to improve in our research. We are trying our best to improve the quality of this manuscript with the suggestions of these nice reviewers.

For your question "the mechanism of interspecific interactions between peanut and maize that leads to the enrichment of these metabolites", we previously found that the belowground signaling cyanide-ethylene are the cues of chemical identification and communication between two different plants (non-legume cassava plant and legume peanut plant) (Chen et al., 2020). These chemical cues simultaneously mediate plant and rhizosphere microbiota interaction. In this study, we focused more on the active response (or active feedback) of legume peanut to diverse plants at the metabolic level. Furey and Tilman (2021) found that long-term diversified cultivation continues to improve soil functioning such as nitrogen regeneration. Our study revealed that the directional regulation of microbial function (such as rhizosphere nitrogen fixation) by leguminous depositable metabolites is

one of the essential reasons to explain the reason for soil functional enhancement. Based on the secondary metabolites identified in this and previous studies (ethylene, flavonoids and coumarins), we speculate that the coexistence of different plant species may stimulate the immunometabolism feedback in plants, which act as the key for plants to “cry for help” from the rhizosphere and establish alliances with soil functional microbiome to adapt to the environment. This suggests that the concept of plant-driven plant-microbiome holobiont needs to be incorporated into future research on the optimization of habitat functions by plant diversification. We’ve added our speculations to the Discussion as follow:

“Interestingly, whether flavonoids and coumarins induced by maize and oilseed rape coexistence or gaseous ethylene which was found to be stimulated by non-legume species recognition¹⁷, these plant secondary metabolites have been reported to not only directly involved in the plant defense of biotic and abiotic stresses but also act as the chemical cue for “belowground cry-for-help”: specific components of root exudates to recruit root-associated and soil-associated microbiomes to enhance plant fitness^{50,51}.”(Line355-360, page 11)

For the figures, we are sorry for the low quality. When uploading revisions, we changed the format of these images. It would be much better now.

References

- [1] Y. Chen, M. Bonkowski, Y. Shen, B. S. Griffiths, Y. Jiang, X. Wang, B. Sun, Root ethylene mediates rhizosphere microbial community reconstruction when chemically detecting cyanide produced by neighbouring plants. *Microbiome* 8, 4 (2020).
- [2] B. Li, Y. Y. Li, H. M. Wu, F. F. Zhang, C. J. Li, X. X. Li, H. Lambers, L. Li, Root exudates drive interspecific facilitation by enhancing nodulation and N₂ fixation. *Proc. Natl. Acad. Sci. U.S.A.* 113, 6496-6501(2016).
- [3] G.N. Furey, D. Tilman. Plant biodiversity and the regeneration of soil fertility. *Natl. Acad. Sci. U.S.A.* 118: e2111321118 (2021).

Major comments:

1-I highly recommend confirming the identification of the metabolites tested (flavonoids and coumarin) using appropriate measurement of a reference standard with MS, MS/MS, and retention time matching.

Re: Yes! Reviewer’s recommendation is correct. In fact, we did use chemical standards and their retention time to match specific compounds that were discovered in the soil. However, due to the number word limitation, we stated the following sentence in the main text:

“Based on primary and secondary mass spectrum analyses and the comparison of standards’ mass spectrum, these four metabolites were putatively identified as quercetin, hyperoside, scopoletin and syringaldehyde (Fig. S4 and S5; Table S2).”

In the revised manuscript, we added a figure of the Mass spectra of four metabolite standards (quercetin, hyperoside, scopoletin and syringaldehyde) using UHPLC–MS/MS as follows. Based on the Fig. S4 and Fig. S5, we identified the specific metabolites.

Fig. S5 Mass spectra of four metabolite standards (quercetin, hyperoside, scopoletin and syringaldehyde) using UHPLC–MS/MS. (A) Extracted ion chromatogram (EIC) of quercetin, hyperoside, scopoletin and syringaldehyde standards; (B) Primary mass spectrum (MS) of the four standards. (C) Secondary mass spectrum (MS/MS) of the four standards.

2-Table S2: change confidence level to mzCloud score. Please add the following information, it is a list of random metabolite names otherwise:

- a. Neutral mass (Da)
- b. Mass Detected ES(-) and/or ES(+)
- c. Relative mass defect (ppm)
- d. RT [min]
- e. Fragment ions

Re: Thank you for your suggestion! We’ve added the information you mentioned in the revised Table S2.

3-Lines 118-119. Why have you set the mzCloud Score Match to 70? The list of compound identification should pass through a manual curation to confirm the identification and confidence level. Please check the standard protocol for compound identification – Schymanski et al, Environmental Science & Technology 2014 48 (4), 2097-2098.

Re: Thanks! We are sorry for the mistakes in our statement “confidence” in the manuscript. In fact, it is the fragment score of Compound Discoverer soft parameter to identify the metabolites using the m/z Cloud database. The related statement has been rephrased to make it clearer (Line 115 and 128, page 4-5). The higher the score, the more accurate metabolite identification. There isn’t a

standard (or threshold) for score cutoff. Taking Liang et al¹ as an example, the MS/MS spectra similarity score cutoff was set as 0.5 (=50%). In our study, we set the score cutoff as 70% after comparing metabolites with different scoring criteria. These metabolites (> 70%) were proper and valuable for later analysis.

The m/z Cloud database, including the MS, MS², RT, etc. information, is a widely used database for metabolomics. Therefore, the MS/MS spectra match was checked to confirm the identifications, which was considered a level 2 identification as the reference article provided by reviewer ².

Here, we used untargeted metabolomics to preliminary screen highly matched specific deposits from thousands of metabolites. Then, for the four key metabolites, we identified them using chemical standards. This part of the experimental results has been added in the revised version. (Line 124-128 Supplementary information) (Fig. S5)

Reference:

[1] L. Liang, M. L. H. Rasmussen, B. Piening, X. Shen, S. Chen, H. Rost, J. K. Snyder, R. Tibshirani, L. Skotte, N. Lee, K. Contrepolis, B. Feenstra, H. Zackriah, M. Snyder, M. Melbye. Metabolic dynamics and prediction of gestational age and time to delivery in pregnant women. *Cell* 181, 1680-1692 (2020).

[2] E. L. Schymanski, J. Jeon, R. Gulde, K. Fenner, M. Ruff, H. P. Singer, J. Hollender. Identifying small molecules via high resolution mass spectrometry: communicating Confidence. *Environ. Sci. Technol.* 48, 2097-2098 (2014).

Minor comments:

line 74: I would change "aboveground diversity" to "plant diversity", as roots of both maize and peanut are part of the belowground.

Re: According to the reviewer's nice suggestion, we changed "aboveground diversity" to "plant diversity". Thank you very much! (line 69, page 3)

Figs: 2B, 3D, 5, S2 are of poor quality, suggest to make them much bigger.

Re: Yes! We are sorry for the poor quality of original Figures. We've saved figures using PDF format in the revised version. That would be much better! (Fig. S4 and S5)

Fig. S2B. Spectra are small and I cannot see the numbers for checking the molecular ions and fragments. Chemical structure is almost transparent.

Re: Sorry again. When uploading revisions, we changed the format of these images. It would be much better now.

Lines 293-295: Provide a reference.

Re: Thank you for your suggestion. We have cited a reference Schmidt et al (2019) which reported that "For instance in soil, typical inter-cell distances of 10–20 μm , and cell-to-cell communication distances of soluble chemicals of up to 78 μm have been described (Ganter et al., 2006). The soil structure and its complex pore space are another driver of cell-to-cell communication and microbial functioning." Based on your suggestion, we listed two more references to support this view. Meanwhile, we changed "...will greatly diminish the diffusion efficiency of these soluble compounds at >10cm distance scale in soil." to "...greatly diminish the diffusion efficiency of these

soluble compounds from neighboring or historical plant species at long distance (such as >10 cm) in soil (Ganter et al., 2006; Schmidt et al., 2019)". In the reference of Schulz-Bohm et al (2019), they stated that 12cm is a long distance of compound diffusion. (Line 289-290, page 9)

References:

[1] R. Schmidt, D. Ulanova, L. Y. Wick, H. B. Bode, P. Garbeva, Microbe-driven chemical ecology: past, present and future. ISME J. 13, 2656-2663 (2019).

[2] S. Gantner, M. Schmid, C. Durr, R. Schuegger, A. Steidle, P. Hutzler, C. Langebartels, L. Eberl, A. Hartmann, F.B. Dazzo FB. In situ quantitation of the spatial scale of calling distances and population density-independent N-acylhomoserine lactone-mediated communication by rhizobacteria colonized on plant roots. FEMs Microbiology Ecology 2006 56:188-194.

[3] K. Schulz-Bohm, S. Gerards, M. Hundscheid, J. Melenhorst, W. de Boer, P. Garbeva. Calling from distance: attraction of soil bacteria by plant root volatiles. ISME J. 12:1252-1262 (2018).

*Lines 352-354. Could you try *Medicago transgenic plants to test the effect of coumarins influence nodulation?**

Re: Thank you for reviewer's nice suggestion. Although supplementing this experiment would take time, we felt it is necessary! Based on your good idea, we grew *Medicago* plants and test the effect of coumarins on root nodulation and plant defense. The details of the experiment design were added below:

“Stimulation of coumarin on the root nodulation of *Medicago truncatula*

The model microorganism *Sinorhizobium meliloti* strain1021 was activated in Yeast extract and Tryptone medium (YT, tryptone 16 g L⁻¹, yeast extract 10 g L⁻¹, NaCl 5 g L⁻¹, pH=7.0). Bacterial cells were washed twice using sterilized water and cell suspensions were adjusted to 0.03 at OD600 for use as microbial agent¹¹.

Seeds of *M. truncatula* (A17) were sterilized using sulfuric acid and then germinated on 1% water agar medium at 4 °C for three days and 25 °C for one day¹¹. Plant seedlings were transplanted to pots (length × width × depth = 10×10×10 cm) with vermiculite for growth (chamber with a 16 h light/8 h dark period at 22 °C; photon flux density = 250 μmol m⁻² s⁻¹). After 7 days, three treatments were set up to compare the effect of coumarin scopoletin on *M. truncatula* nodulation: (1) SM, seedling was injected with 5 mL microbial agent; (2) SML, seedling was injected with 5 mL microbial agent containing 5 μg mL⁻¹ scopoletin; (3) SMH, seedling was injected with 5 mL microbial agent containing 50 μg mL⁻¹ scopoletin. Each treatment was conducted 12 replicates. Controls with water injection were processed identically. Pots were placed back in the chamber. After 3 days of inoculation, 6 plant replicates of each treatment were collected for root RNA extraction and qRT-PCR determination. Other 6 replicates were collected for root nodule calculation and biomass detection after 30 days of inoculation. During plant growth, pots were sprayed with water every 5 days.” (Line 147-165, supplementary materials)

Thanks for the assistance of Prof. Ertao Wang's team who has a fairly mature technique to study the progress of model *Medicago* nodulation (Dong et al., 2020). We obtained the following results after one month:

Fig. S10 Effect of coumarin scopoletin on the root gene expression of *Medicago truncatula*. (A) Genes involved in root nodule formation and host defense. (B) Photograph of plant growth and plant belowground traits on the 30th day. The data are shown as the mean \pm standard deviation ($n=6$). The error bars with lowercase indicate significant differences between groups ($P<0.05$) via one-way ANOVA and Tukey's post-hoc tests. C, control; SM, seedling inoculated with rhizobia *Sinorhizobium meliloti*; SML, seedling inoculated with rhizobia *S. meliloti* strain1021 and low concentration ($5 \mu\text{g mL}^{-1}$) of scopoletin; SMH, seedling inoculated with rhizobia *S. meliloti* strain1021 and high concentration ($50 \mu\text{g mL}^{-1}$) of scopoletin.

Obviously, “the addition of trace scopoletin ($5\mu\text{g mL}^{-1}$) both upregulated genes involved in root nodulation (including *MtCCaMK*, *MtNIN*, *MtERN1*, *MtVAPYRIN*, *MtENOD11*, *MtRIP1* and *MtFLOT4*) and plant defense (including *MtPR4*, *MtPR10* and *MtGST*) ($P<0.05$), which finally lead to an increase in root nodules and biomass of Madicago compared with control (C) and single bacterial inoculation (SM). Comparatively, the addition of high scopoletin ($50\mu\text{g mL}^{-1}$) weakens the effect of plant nodulation and defense compared with low scopoletin addition.” (Line 344-350, page 11)

References:

[1] W. T. Dong, Y. Y. Zhu, H. Z. Chang et al., An SHR-SCR module specifies legume cortical cell fate to enable nodulation. *Nature* 589: 586 (2020).

Methods, line 443: extracted in methanol or in acetonitrile (as describe in the supplemental material)?

Re: Oh! We are very sorry for our mistake! It's “extracted in acetonitrile”. We've corrected it in the main text. (Line 460, page 14)

Reviewers' Comments:

Reviewer #1:

Remarks to the Author:

The authors have done a great job to response my comments. I now only have several minor suggestions.

Line 134 change "(scopoletin)"and "(syringaldehyde)" to (eg. scopoletin) and (eg. syringaldehyde)

Line 228 Reference 22 and 23 are not relevant to this sentence. I think it is reference 21.

Line 715-720 Reference 30 and 31 are not relevant to the revised main text. Delete them.

Supplementary Figures

"Table S9" in the legend of Fig.S8 is wrong. It must be Table S8?

Fig.S6 For the gene expression, please check the data show as " $\log_2(\Delta\Delta CT)$ " or $2^{-\Delta\Delta CT}$?

Supplementary Tables

The title "Table S9 Information of enriched ASVs in the peanut rhizosphere of different cropping systems" should be change to "Table S7 Information of enriched ASVs in the peanut rhizosphere of different cropping systems"

The title "Table S7 Information on four enriched ASVs in the most diverse cropping system and specific bacterial isolates" should be change to "Table S9 Information of four enriched ASVs in the most diverse cropping system and specific bacterial isolates"

Reviewer #2:

Remarks to the Author:

Firstly, I want to thank the authors for the tremendous effort they have put into updating this publication.

However, I still cannot accept this publication because I have been able to confirm my suspicions, and this experimental design features pseudoreplicates, not true replicates.

The authors collected samples within each plot, so these samples are not independent as they depend on a new factor: the plot. Therefore, they cannot be considered as 9 replicates since this dataset has 3 replicates and 6 pseudoreplicates.

The primary assumption of one of the most commonly used techniques, ANOVA, is the independence of observations, which is not met here due to the samples depending on the plot.

Therefore, the results of most statistical analyses lack meaning as the assumptions are not met and we cannot trust in the results and conclusions.

The error lies in the sampling design, as there are more samples than real plots, and these samples are dependent of the plot. To maintain true replicates, one could either work with a single sample per plot or calculate the mean of all pseudoreplicates within each plot to ensure a random selection. Another option could be to repeat the experiment with a higher number of experimental plots.

To test the hypothesis, the authors must ensure that the samples (observations) are statistically independent.

Reviewer #4:

Remarks to the Author:

In this manuscript, the authors demonstrated that the neighboring maize plants affect the rhizosphere metabolites of peanut, which could lead to the altered microbiome that enhances nitrogen fixation. The authors addressed most of the reviewers' comments, especially regarding the identification of rhizosphere metabolites. Although this study demonstrated the influence of neighboring maize plants on peanut growth, the molecular mechanisms underlying these effects are still not clearly understood. I have several comments that need to be addressed.

Flavonoids and coumarins were identified using authentic standards. I recommend quantifying these metabolites present in the rhizosphere of peanuts from both PP and PM-R fields. The mechanisms behind the effects of these metabolites on the rhizosphere microbiota and their functions can be explored. For instance, research can be conducted to investigate whether these metabolites act as chemoattractants or induce gene expression of nod genes at concentrations present in the rhizosphere.

This is a long-term (8 years) crop rotation study, but the peanut growth data were not fully presented in this manuscript. As the authors responded to the reviewer's comment, a number of peanuts developed disease. Does this happen every year or just this time? It would be nice if the authors would show the growth data in the Supplementary Materials.

As the authors discussed the role of flavonoids and coumarins as either carbon sources or molecular signals, it is crucial to unveil the molecular mechanisms of this phenomenon. To address this issue, I suggest the authors conduct tests to determine whether isolated bacteria can utilize these metabolites as a carbon source and determine whether they impact expression, particularly nod genes.

I am uncertain about the suitability of using Medicago plants. Do the symbiotic rhizobia of peanuts also form nodules in Medicago? While nod-gene inducing flavonoids of *Sinorhizobium meliloti* are well-documented, I am curious whether the flavonoids and coumarins found in the peanuts rhizosphere can also induce the nod genes.

Point-by-point responses to comments from Reviewers

(Legume rhizodeposition of flavonoids and coumarin promotes nitrogen fixation by soil microbiota under crop diversification, Manuscript ID: NCOMMS-23-15056A)

Reviewer #1 (Remarks to the Author):

The authors have done a great job to response my comments. I now only have several minor suggestions.

Response: We appreciate the reviewer's acknowledgment of our efforts in enhancing the manuscript. We have further enhanced the manuscript in response to the second-round comments provided below.

Line 134 change "(scopoletin)" and "(syringaldehyde)" to (eg. scopoletin) and (eg. syringaldehyde)

Response: OK! Done. (Line 133-134, page 5)

Line 228 Reference 22 and 23 are not relevant to this sentence. I think it is reference 21.

Response: We have excluded additional references (21, 22 and 23) as the review paper [reference 6] adequately supports the point made in this sentence. (Line 228, page 7)

Line 715-720 Reference 30 and 31 are not relevant to the revised main text. Delete them.

Response: The two references have been deleted.

Supplementary Figures

"Table S9" in the legend of Fig.S8 is wrong. It must be Table S8?

Response: Done. (Table S8)

Fig.S6 For the gene expression, please check the data show as "log₂ ($\Delta\Delta CT$)" or $2^{-\Delta\Delta CT}$?

Response: Thanks for reviewer's reminder. We checked our protocol and yes the data should be $2^{-\Delta\Delta CT}$. Sorry for the mistake! We have corrected the formula. (Line 93, Supplementary information)

Supplementary Tables

The title "Table S9 Information of enriched ASVs in the peanut rhizosphere of different cropping systems" should be change to "Table S7 Information of enriched ASVs in the peanut rhizosphere of different cropping systems"

Response: Done. (Table S7)

The title "Table S7 Information on four enriched ASVs in the most diverse cropping system and specific bacterial isolates" should be change to "Table S9 Information of four enriched ASVs in the most diverse cropping system and specific bacterial isolates"

Response: Done. (Table S9)

Reviewer #2 (Remarks to the Author):

Firstly, I want to thank the authors for the tremendous effort they have put into updating this publication.

Response: Thanks for the reviewer's recognition of our effort in improving the manuscript.

However, I still cannot accept this publication because I have been able to confirm my suspicions, and this experimental design features pseudoreplicates, not true replicates. The authors collected samples within each plot, so these samples are not independent as they depend on a new factor: the plot. Therefore, they cannot be considered as 9 replicates since this dataset has 3 replicates and 6 pseudoreplicates. The primary assumption of one of the most commonly used techniques, ANOVA, is the independence of observations, which is not met here due to the samples depending on the plot. Therefore, the results of most statistical analyses lack meaning as the assumptions are not met and we cannot trust in the results and conclusions. The error lies in the sampling design, as there are more samples than real plots, and these samples are

dependent of the plot. To maintain true replicates, one could either work with a single sample per plot or calculate the mean of all pseudoreplicates within each plot to ensure a random selection. Another option could be to repeat the experiment with a higher number of experimental plots.

To test the hypothesis, the authors must ensure that the samples (observations) are statistically independent.

Response: In response to the reviewer's significant concerns regarding sampling duplication, we have thoroughly addressed this issue through a detailed discussion with all co-authors. In alignment with the reviewer's suggestion, we calculated the mean of all so-called pseudoreplicates within each plot to ensure a random selection. Therefore, the independent replicates from the field experiments were reduced from nine to three. Then we did the ANOVA statistical analysis based on the three replicates. We also added the following sentence in "Statistical analyses":

"For field experiment data, we calculated the mean of all samples (n=3) for each plot to ensure a random selection for ANOVA analysis. Therefore, the replicates for each field treatment were three instead of nine." (Line 584-586, page 18)

To elucidate the origin of these average data, we shaded the original duplicate points as grey, presenting them as background elements in main figures as follow. In general, we observed no alteration in the trends of the data, and the significance became more pronounced in the ANOVA statistical analysis after using the mean of all pseudoreplicates.

“Figure 1 Effect of crop diversification on peanut growth, root nodulation and rhizosphere N availability. **(A)** Effect on peanut nodulation. **(B)** Effect on peanut rhizosphere ¹⁵N fixation. The data in A-B are shown as the mean \pm standard deviation (n=3). The error bars with lowercase indicate significant differences between groups (p<0.05) via one-way ANOVA and Tukey’s post-hoc tests. **(C)** Correlations between root nodulation and peanut plant biomass. **(D)** Correlations between soil ¹⁵N fixation and plant biomass. **(E)** Correlations between root nodulation and rhizosphere nitrogen components including total (black points with regression), ammonium (blue points with regression) and nitrate (green points with regression) nitrogen. **(F)** Correlations between soil ¹⁵N fixation and rhizosphere nitrogen components including total, ammonium and nitrate nitrogen. Lines represent the least squares regression fits and shaded areas represent the 95% confidence intervals. Original value of each sample was marked with gray dot in the background. PP, P-R and PM-R represent peanut monocropping, peanut-oilseed rape rotation, and peanut-maize intercropping rotated with oilseed rape, respectively.”

“Figure 2A-C Effect of crop diversification on metabolic production in the peanut rhizosphere. (A) Principal component analysis of the metabolites detected in peanut rhizosphere soil. QC, quality control samples (composed of a small aliquot of each sample). 95% confidence ellipses are shown around each group to distinguish community differences (n=7). (B) Screening for specific enriched metabolites (m/z Cloud database best match >95%). Four metabolites, identified as quercetin, hyperoside, scopoletin and syringaldehyde, were selected based on the fold change (>2) of relative concentration (PM-R vs P-R, and PM-R vs PP) by t-test (n=3). (C) Principal component analysis of root transcriptomic variance of PP and PM-R. 95% confidence ellipses are shown around each group to distinguish community differences (n=7). Original value of each sample was marked with gray dot in the background. PP, P-R and PM-R represent peanut monocropping, peanut-oilseed rape rotation, and peanut-maize intercropping rotated with oilseed rape, respectively.”

“Figure 3A-B Effect of crop diversification on the peanut rhizosphere bacterial community. (A) Shannon and Chao1 richness indices. The error bar with lowercase indicates significant difference between groups (p<0.05) via one-way ANOVA and Tukey’s post-hoc tests (n=3). (B) Principal coordinate analysis (PCoA) of bacterial beta dispersion among different samples based on Bray–Curtis distance (left) and distance of centroid beta-dispersal values for groups (right). Black lines indicate the median values. P values were adjusted using multiple (95% family-wise confidence level) comparisons using Tukey’s HSD. Original value of each sample was marked with gray dot in the background.”

“Fig.S2A Effect of crop diversification on peanut performance. (A) Effect on peanut plant height, biomass and fruit weight and N uptake (n=3). Original value of each sample was marked with gray dot in the background. The error bars with lowercase indicate significant differences between groups (p<0.05) via one-way ANOVA and Tukey’s post-hoc tests. Original value of each sample was marked with gray dot in the background. PP, P-R and PM-R represent peanut monocropping, peanut-oilseed rape rotation, and peanut-maize intercropping rotated with oilseed rape, respectively.”

“**Fig.S6B** Validation of representative up- and down-regulated gene expressions by real-time polymerase chain reaction. Gene IDs marked with pink in A were selected for validation. Data are shown as $2^{-\Delta\Delta C_t}$ and presented as the mean \pm standard deviation ($n=3$). The error bars with numbers represent significant differences (p values) as determined by t-test according to F test results. PP, peanut roots from peanut monocropping; PM-R, peanut roots from peanut maize intercropping rotated with oilseed rape.”

Reviewer #4 (Remarks to the Author):

In this manuscript, the authors demonstrated that the neighboring maize plants affect the rhizosphere metabolites of peanut, which could lead to the altered microbiome that enhances nitrogen fixation. The authors addressed most of the reviewers' comments, especially regarding the identification of rhizosphere metabolites. Although this study demonstrated the influence of neighboring maize plants on peanut growth, the molecular mechanisms underlying these effects are still not clearly understood. I have several comments that need to be addressed.

Response: Thanks for the reviewer’s recognition of our effort in improving the manuscript based on reviewer 3’s comments. We would also like to express our gratitude for reviewer 4’s valuable time spent on reviewing our paper. In consideration of your comments on the molecular mechanism details mentioned below, we have made additional improvements to the paper and addressed each comment individually in our responses.

Flavonoids and coumarins were identified using authentic standards. I recommend quantifying these metabolites present in the rhizosphere of peanuts from both PP and PM-R fields. The mechanisms behind the effects of these metabolites on the rhizosphere microbiota and their functions can be explored. For instance, research can be conducted to investigate whether these metabolites act as chemoattractants or induce gene expression of nod genes at concentrations present in the rhizosphere.

Response: In the first stage of our research, we used soil non-targeted metabolism to figure out the specific metabolites of peanut that induced by the crop diversification. Then, we used targeted metabolism and a series of microbial assays to answer whether these specific metabolites have the ability to combine with microorganisms to enhance the environmental adaptability of host plant peanut (taking nitrogen acquisition

as the example). Based on target metabolism validation, we found that the concentration of flavonoids and coumarins were within the range of 0.01-20 $\mu\text{g mL}^{-1}$. Therefore, we choose $5\mu\text{g mL}^{-1}$ for the microplate inoculation (Fig. 4) and peanut plant inoculation culture (Fig. 5). In response to your comment, we added this information in the main text “On average, these individual metabolites were found to affect the growth rate (V), either positively or negatively, of 48% of the tested strains in 1/5 TSB medium containing $5\mu\text{g mL}^{-1}$ one of different metabolites (Fig. 4B, t-test, $p<0.05$).” (Line 212-215, Page 7)

We found that these metabolites have different regulatory mechanisms on different nitrogen-fixing bacteria. For free-living N_2 -fixers, metabolic addition stimulates the growth of them and improving their N_2 -fixing ability (Fig. 4B-C), indicating these N_2 -fixers may use flavonoids and coumarins as carbon resources.

“**Figure 4** Bacterial isolates from the PM-R peanut rhizosphere: effects of typical metabolites on bacterial growth rates, free-living N_2 fixation. **(B)** Effect of rhizosphere metabolites on the growth rate (V) of selected bacterial strains, measured in microplate assays during the bacterial logarithmic growth phase (n=6). $V>1$ represents growth promotion with metabolite addition; $V<1$ represents growth inhibition. Bars with asterisks represent significant differences as determined by t-test (* $p<0.05$, ** $p<0.01$, *** $p<0.001$). **(C)** Correlations between bacterial growth of free-living N_2 fixers and their capability for N_2 fixation. Lines represent the least squares regression fits and shaded areas represent the 95% confidence intervals.”

For symbiotic N_2 -fixers, flavonoids and coumarins do not stimulate microbial growth, but induce microbial colonization and host nodulation. As shown in Fig. 5 as follow, flavonoids (quercetin and hyperoside) and coumarins (Scopoletin) activate the nod gene (*nodC* and *nodD1*) of peanut Bradyrhizobium (N47) and nodulation genes *AhSYM*RK, *AhCCa*MK and *AhNIN* in peanut roots (Fig.5).

Figure 5 Bacterial isolates from the PM-R peanut rhizosphere: effects of typical metabolites on Bradyrhizobium N47 colonization. (A) Diagram of symbiosis signaling pathway of Bradyrhizobium N47 in peanut root, inducing nodule formation at lateral root bases. (B) Expression of bradyrhizobial nodulation signaling genes, 48h after addition of metabolites. (C) Expression of host common symbiotic signaling genes, measured in peanut roots 24h after bacterial inoculation and metabolite addition. (D) Peanut root nodule number 30 days after bacterial inoculation and metabolite addition. The error bars with lowercase represent significant differences between groups ($p < 0.05$) via one-way ANOVA and Tukey's post-hoc tests. C, control; Br, Bradyrhizobium; Qu, quercetin; Hy, hyperoside; Sc, scopoletin; Sy, syringaldehyde."

Based on these results, we made the following conclusion: for free-living N_2 -fixers, specific metabolites at least were carbon resources for microbial growth promotion; for symbiotic N_2 -fixers, specific metabolites acted as chemical signals to assist in microbial colonization and host nodulation. The question of Whether and how these metabolites act as chemoattractants for specific bacteria is truly intriguing and warrants exploration in our future research. Our team is currently engaged in chemoattraction verification work, and we look forward to sharing our findings through publication in the future. Thank you!

This is a long-term (8 years) crop rotation study, but the peanut growth data were not fully presented in this manuscript. As the authors responded to the reviewer's comment, a number of peanuts developed disease. Does this happen every year or just this time? It would be nice if the authors would show the growth data in the Supplementary Materials.

Response: The data concerning the growth of peanuts in 2019 were presented in Fig. S1. The original focus of the field experiment was to observe changes in soil fertility, with early scientists placing greater emphasis on planting and variations in soil nutrients rather than on plant disease.

We observed the occurrence of peanut diseases in 2019, and conducted preliminary disease statistics from 2023 (Due to the Covid-19 pandemic, data collection [2020-2022] for field experiments has been impacted). It still takes time to confirm the stability of the results. The peanut biomass data from 2023 are consistent with the results of this study. We wish it could be published alongside other results later. In this context, histograms were shown to illustrate the data trends in 2023. Overall, the observed data trends are similar to those in 2019.

Fig.1 The effect of different crop diversification on peanut plant growth in 2023. The error bars with lowercase represent significant differences between groups ($p < 0.05$) via one-way ANOVA and Tukey's post-hoc tests.

As the authors discussed the role of flavonoids and coumarins as either carbon sources or molecular signals, it is crucial to unveil the molecular mechanisms of this phenomenon. To address this issue, I suggest the authors conduct tests to determine whether isolated bacteria can utilize these metabolites as a carbon source and determine whether they impact expression, particularly *nod* genes.

Response: Thanks for reviewer's suggestion. Our results actually supported that flavonoids and coumarins can serve either as carbon sources or as molecular signals through bacterial isolation incubation experiment. The specific role of these metabolites is contingent upon the functional traits of bacterial N_2 -fixers. In the case of free living N_2 -fixers, their biomass increased upon the addition of metabolites, suggesting free living N_2 -fixers can use metabolites as carbon resources. In contrast to free living N_2 -fixers, the growth of peanut symbiotic N_2 -fixers was not promoted with the addition of metabolites. Instead of it, their survival strategies changes, evidenced by the up-regulation of *nod* genes, including *nod D* and *nod C1* in response to metabolic addition. Meanwhile, alternations were observed in root genes associated with peanut nodulation initiation, indicating that flavonoids and coumarins act as signals to participate in plant root-functional microbial interactions. We're sorry that our descriptions and instructions in the Discussion may have caused misunderstandings with the reviewer. Therefore, we revised the sentence to "In the case of these N_2 -fixers, specific flavonoids and coumarin more like carbon resources, supporting microbial survival and nitrogen fixation." (Line 320-322, page 10)

I am uncertain about the suitability of using *Medicago* plants. Do the symbiotic rhizobia of peanuts also form nodules in *Medicago*? While *nod*-gene inducing flavonoids of *Sinorhizobium meliloti* are well-documented, I am curious whether the flavonoids and coumarins found in the peanuts rhizosphere can also induce the *nod* genes.

Response: During the first-round of review, we interpreted reviewer 3's question as inquiring about whether diversification-induced metabolites are universally effective in stimulating the interaction between leguminous plants and rhizobia. It's important to note that the peanut is not the model plant for legume-rhizobia interaction studies; instead, *Medicago* serves as the model. Both flavonoids and coumarins have reported to play crucial roles in plant root-microbial interactions. Flavonoids have been identified as regulators of *nod* genes. If coumarins can be demonstrated to enhance the interaction between a model legume and its rhizobia as well, it has the potential to attract more researchers to engage in related studies, significantly expanding our current understanding of the signals involved in legume-rhizobia interactions. Inspired by reviewer 3's comments, we have supplemented the experiment with *Medicago* plants.

To assess the impact of coumarins on nodulation initiation, we used *Medicago truncatula* (A17) along with its root-associated bacteria, *Sinorhizobium meliloti* strain1021, instead of peanut *Bradyrhizobium* N47. The symbiotic ability of *S. meliloti* strain1021 in *M. truncatula* A17 has been well-documented (Dong et al., 2020). Further details of this inoculation and culture experiment are described in "Stimulation of coumarin on the root nodulation of *Medicago truncatula*" section. (Line 147-165, supplementary information)

To investigate whether the flavonoids and coumarins present in the peanut rhizosphere could induce the expression of *nod* genes, we present our results as follows:

"Therefore, we focused on this strain to assess whether the enriched root metabolites increased nodulation signaling during the establishment of the symbiosis. Indeed, the addition of flavonoids and coumarin to pure *Bradyrhizobium* cultures increased the expression of *Bradyrhizobium nodD1* and *nodC* genes by 21-126%

(*NodD1*) and 216-430% (*NodC*), compared to controls (Fig. 5B). Simultaneously, the addition of these metabolites to peanut seedlings inoculated with *Bradyrhizobium* N47 enhanced peanut root gene expression of *AhSYMRK*, *AhCCaMK* and *AhNIN* by 43-169% at the transcriptional level compared to *Bradyrhizobium* inoculation without added metabolites (Fig. 5C). These upregulated plant genes play crucial roles in nodule organogenesis by encoding leucine-rich repeat receptor-like kinase (*SYMRK*), calcium spikes by a calcium calmodulin-dependent protein kinase (*CCaMK*) and an RWP-RK transcription factor (*NIN*)^{31,32}.” (Line 236-247, Page 8)

References:

[1] W. T. Dong, Y. Y. Zhu, H. Z. Chang et al., An SHR-SCR module specifies legume cortical cell fate to enable nodulation. *Nature* 589: 586 (2020).

Reviewers' Comments:

Reviewer #2:

Remarks to the Author:

The authors calculate the average to avoid the pseudo-replicates. The paper is ok for publication

Reviewer #4:

Remarks to the Author:

I appreciate the authors for addressing the comments. Although most of the answers are satisfactory, I still wonder if flavonoids and coumarins found in peanuts induce the nod genes of *S. meliloti*. It is highly recommended to test if these compounds induce the nod genes as shown for *Bradyrhizobium*. Without these data, we cannot argue for the same mode of action for peanut and *Medicago*.

Point-by-point responses to comments from Reviewers

(Legume rhizodeposition promotes nitrogen fixation by soil microbiota under crop diversification, Manuscript ID: NCOMMS-23-15056B)

Reviewer #2 (Remarks to the Author):

The authors calculate the average to avoid the pseudo-replicates. The paper is ok for publication

Response: We would like to express our sincere gratitude for your final approval of our paper. Your valuable time and the multitude of insightful suggestions you provided have played a crucial role in significantly enhancing the quality of our work. We truly appreciate your dedication and expertise in reviewing our paper.

Reviewer #4 (Remarks to the Author):

*I appreciate the authors for addressing the comments. Although most of the answers are satisfactory, I still wonder if flavonoids and coumarins found in peanuts induce the nod genes of *S. meliloti*. It is highly recommended to test if these compounds induce the nod genes as shown for *Bradyrhizobium*. Without these data, we cannot argue for the same mode of action for peanut and *Medicago*.*

Response: Thank you for acknowledging the revisions made to our paper. In response to your recommendation, we recognize the significance of providing evidence that universal patterns of interaction between specific compounds and microorganisms. Consequently, we have conducted additional experiments for Nod gene test.

The results demonstrate that there is a noticeable stimulation of nod genes expression in *S. meliloti* when culture contains $0.1\mu\text{g mL}^{-1}$ of metabolites (as illustrated in the figure below). We have included the details in the supplementary figure (Figure S10). Additionally, a brief explanation of the experiment and the result have been incorporated into the main text and supplementary materials.

“Similar stimulating effects on the nod genes (*NodDI* and *NodC*) have also been observed in the model microorganism *Sinorhizobium meliloti* (strain1021) for such metabolites (Fig. S10).” (Line 240-242, page 8 main text)

“To investigate whether the rhizosphere metabolites influenced the nodulation signaling of *S. meliloti* strain1021, we measured the expression of bacterial nod genes in this strain after metabolite exposure. Briefly, 100 μl of bacterial suspension ($\text{OD}_{600}=0.5$) was transferred to 5 mL YT media in culture tubes and incubated at 28°C with shaking at 200 rpm for 2 h. Then, each culture tube received one of the four metabolites (Qu, Hy, Sc, Sy) to a final concentration of $0.1\mu\text{g mL}^{-1}$. The mixtures were cultured under the same conditions for another 12 h. The bacterial cells were collected for RNA extraction and qRT-PCR of Nod genes (Supplementary Materials). Controls (C) of *S. meliloti* with water addition were processed identically. Each treatment group included six replicates. ” (Line 152-161, supplementary information)

Fig. S10 Expression of *Sinorhizobium meliloti* strain1021 Nod genes, measured 10h after addition of metabolites.

We hope these modifications address your concerns and contribute to the overall clarity and robustness of our findings.

Reviewers' Comments:

Reviewer #4:

Remarks to the Author:

I appreciate the authors for addressing the comment.